# DisIR: Disentangled Learning of Controllable All-in-One Image Restoration under Composite Degradations

## Abstract

Composite degradation scenarios, in which multiple types of degradation are mixed together, have attracted increasing interest in the development of restoration models. Although prior knowledge of degradation types exists, the challenge of precise image restoration persists, particularly when multiple degradations are intricately mixed, and selectively handling individual degradations poses considerable difficulty. To tackle this challenge, we propose DisIR, a novel disentangled framework that learns controllable representations for composite image restoration through four distinct training objectives. First, we introduce an identity embedding as a prompt, along with an identity loss that guides the model to reproduce the input without modification. Second, we design a ratio control mechanism where the identity embedding can be linearly combined with degradation-specific embeddings at controllable ratios, enabling fine-grained restoration intensity control through a dedicated ratio control loss. Third, to disentangle multiple degradations, we incorporate an intermediate loss that supervises intermediate outputs, each aimed at selectively removing only one type of degradation among multiple composite degradations. Fourth, a permutation-invariant loss is applied to enforce consistent restoration results, regardless of the order in which multiple degradations are removed. By focusing on the training pipeline, our method acts as a versatile enhancement that can be integrated into controllable architectures without requiring their structural redesign. Experimental results demonstrate that our DisIR achieves state-of-the-art performance on composite degradation benchmarks while enabling flexible and selective removal of multiple degradations, either sequentially or in a single step, through a fused embedding with user-controlled intensity ratios.

## 1 Introduction

In the field of low-level computer vision, the key problem of image restoration is dedicated to rebuilding high-quality images from sources affected by various adverse degradations. This task is critical in applications, such as autonomous driving (Shyam & Yoo, 2024; Lin et al., 2025) and robotics (Porav et al., 2019), where maintaining reliable visual input is inherently difficult. Despite its importance, image restoration is inherently ill-posed, as there can be infinitely many valid solutions, making generalization across diverse conditions challenging. Earlier methods (He et al., 2010; Qu et al., 2017; Zhang & Patel, 2018) used mainly hand-designed features tailored for particular types of degradation. Later, attention-based models such as Uformer (Wang et al., 2022) and Restormer (Zamir et al., 2022a) achieved strong results in restoration tasks using transformer architectures (Vaswani et al., 2017). Although these approaches work well for single degradations, they typically handle only one degradation type per model, requiring multiple specialized networks and reducing flexibility for diverse restoration tasks.

In response to these shortcomings, the all-in-one restoration paradigm proposes a single model capable of handling a wide range of restoration tasks. Representative approaches include corruption-agnostic models like AirNet (Li et al., 2022), weather-focused architectures such as WGWS (Zhu et al., 2023) and TKL (Chen et al., 2022b), and masked pre-training methods like RAM (Qin et al., 2024). In particular, prompt-based models such as PromptIR (Potlapalli et al., 2023) have emerged, using learned embeddings to guide the restoration process adaptively. However, despite these advances,

most all-in-one restoration studies have focused on treating the types of degradation individually, rather than handling them in complex composite settings. Recognizing this limitation, composite degradation models such as OneRestore (Guo et al., 2024) have been developed to tackle scenarios where multiple degradations are mixed. Although OneRestore is a significant contribution, it faces notable challenges in separating and removing these blended corruptions effectively. In particular, we observe that OneRestore suffers from limited controllability. (1) It lacks an explicit mechanism to bypass the input and preserve the original content. While a prompt like "clear" can keep clean inputs unchanged, users have no specific way to tell the model to leave degradations in place or to do nothing at all. (2) It also does not support soft removal of degradations. For example, when a user wants to retain a certain degree of haze for visual effect, the model does not offer a way to control the extent of removal. Controllability issues become more pronounced in complex degradation scenarios. (3) Selectively removing one degradation from a composite image often leads to the introduction of new, unintended artifacts (4) Furthermore, the restoration result is highly sensitive to the order in which degradations are removed. For example, removing haze first and then enhancing low-light yields a noticeably different result compared to enhancing low-light first and then removing haze. These issues arise mainly from the limited controllability of previous approaches, which becomes a significant limitation when multiple degradations are handled.

To address these challenges and further expand controllability, we propose Disentangled Image Restoration (DisIR) for composite degradations with disentangled learning, a novel framework that enhances the adaptability and effectiveness of all-in-one restoration models in composite scenarios. Our approach introduces the following components to enhance controllability and effectiveness in image restoration: (1) Identity embedding and identity loss, which guide the model to preserve the input when no restoration is needed, thereby preventing unnecessary modifications. (2) A ratio control loss, which enforces proportional degradation removal by using a linear combination of identity embedding and degradation-specific embeddings at a controllable ratio. (3) An intermediate loss that supervises partially restored outputs, encouraging the model to learn fine-grained restoration steps and improve overall performance. (4) A permutation-invariant loss that ensures consistent restoration results regardless of the order in which degradations are removed. These components are integrated into a comprehensive disentangled restoration pipeline that enables structured, adaptive, and controllable restoration across diverse degradation scenarios. The proposed model can remove multiple degradations either sequentially or in a single step, while providing users with flexible control over the removal intensity of each degradation. Experimental results on benchmark datasets for composite degradations (Guo et al., 2024) show that our DisIR achieves state-of-the-art restoration performance while providing superior controllability. This capability is further supported by results from ablation studies.

## 2 RELATED WORK

Image restoration encompasses various low-level vision tasks, including dehazing (He et al., 2009; Chen et al., 2016; Zheng et al., 2023), deraining (Kang et al., 2012; Fu et al., 2017; Chen et al., 2024), desnowing (Dalal & Triggs, 2005; Bossu et al., 2011; Quan et al., 2023), and low-light enhancement (Land, 1977; Wang et al., 2013a; Zhou et al., 2023b). While the aforementioned methods perform well on individual or at most two degradations, they require separate models for different tasks. This paper explores a unified model that handles multiple degradations simultaneously while allowing selective removal of specific ones.

**All-in-One Image Restoration.** Recent developments in image restoration have explored various techniques to handle images affected by multiple types of degradation. One prominent approach is the partial parameter-sharing One-to-Many strategy, which utilizes a shared backbone with multiple input and output pathways to address different degradation factors (Li et al., 2020; Chen et al., 2021a; Wang et al., 2023b). Beyond this, recent efforts have shifted toward fully shared-weight architectures for universal restoration, enabling more flexible solutions. For instance, Chen et al. (2022b) proposed a unified model for adverse weather removal using a two-stage knowledge learning framework, while Li et al. (2022) introduced AirNet, which handles unknown corruptions without prior knowledge. Özdenizci & Legenstein (2023) further explored a diffusion-based approach for restoring patch-wise degradations across diverse conditions. However, as existing universal methods struggle with interference between different degradation types in complex scenarios, prompt-based approaches are emerging to adaptively guide restoration across multiple degradations. PromptIR (Pot-

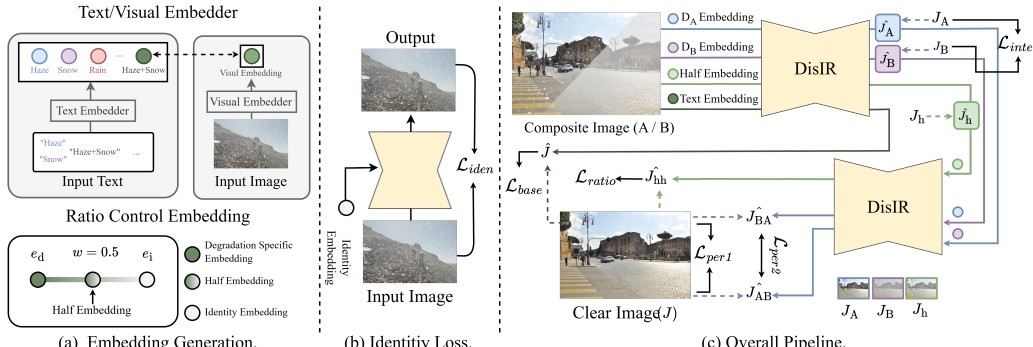

Figure 1: **Overview of our DisIR.** (a) Embedding Generation shows the construction of degradation-specific embeddings through learned Visual/Text embedders and the ratio control mechanism. Ratio Control Embedding enables continuous control of restoration intensity by interpolating between our defined identity embedding and degradation embeddings. (b) The identity loss is a loss function designed specifically for the identity input, which preserves the input without any modifications. It serves to suppress changes in the model's output when the model receives an identity embedding as input. (c) Overall Pipeline of our DisIR. The restoration model handles composite images with multiple degradations ($D_A$ and $D_B$ simultaneously, denoted as $D_{AB}$) using embeddings selected from pre-trained Visual/Text Embedder sets (a). The Half Embedding is formed by weighted combination of Identity and $D_{AB}$ embeddings, and each restoration flow is processed following the guidance of its corresponding same-colored pipeline

lapalli et al., 2023) is a prompt-based learning approach for all-in-one image restoration that encodes degradation-specific information to dynamically guide restoration. Guo et al. (2024) recently introduced OneRestore, which incorporates prompt-based control in image restoration by utilizing versatile scene descriptors to enable adaptive and controllable restoration for diverse composite degradations. While OneRestore provides a unified framework, it cannot perform selective degradation removal and suffers from order dependency in restoration sequences. To address these limitations and enhance fine-grained controllability, we propose a DisIR restoration framework with disentangled learning and tailored novel loss functions.

## 3 METHOD

We introduce a disentangled learning pipeline for selective and ratio-controlled degradation removal, based on four novel losses in Fig. 1. First, we initialize our model using the architecture and training strategy of OneRestore (Guo et al., 2024), then proceed to the disentangled learning pipeline once convergence is reached. Subsequently, we jointly optimize the model with all proposed loss functions, including identity, ratio control, intermediate, and permutation-invariant loss functions, along with the baseline loss defined in the original training setup.

### 3.1 BASELINE

Inspired by OneRestore, our baseline model adopts an encoder-decoder architecture with a Scene Descriptor-guided Transformer Block (SDTB) at its core to handle composite image degradations. To provide degradation-specific guidance, we incorporate a separate Text/Visual embedder that generates scene descriptors representing the types of degradations present in the input. These descriptors are injected into the SDTB via Scene Descriptor-guided Cross-Attention (SDCA), enabling the model to focus on the degradations to be removed. During training, the Text/Visual embedder is first trained using a cosine cross-entropy loss. Subsequently, the baseline restoration model is trained with the pre-trained embedder, using a combination of smooth L1 loss, MS-SSIM loss $\mathcal{L}_{\text{L1-SSIM}}$, and composite degradation restoration loss $\mathcal{L}_{\text{c}}$, as follows:

$$\mathcal{L}_{\text{base}} = \mathcal{L}_{\text{L1-SSIM}}(J, \hat{J}) + \alpha \mathcal{L}_{\text{c}}(J, \hat{J}, I, \{I_o\}), \qquad (1)$$

where $J$ and $\hat{J}$ denote the ground truth (GT) and model output, $I$ is the input image, $I_0$ is the set of negative examples (i.e., images in the batch with degradations different from the current restoration target), and $\alpha$ is a hyperparameter controlling the relative weights of the two loss terms.

## 3.2 IDENTITY EMBEDDING

The identity operation refers to the case where the model outputs the input image without any modifications. While the baseline Text/Visual embedder is designed for restoration tasks, it does not account for the identity operation. To support this, we assign a dedicated embedding vector that explicitly represents the identity operation. This embedding vector is predefined as a constant filled with ones and does not require any additional training. We refer to this constant vector as the identity embedding. The baseline restoration model is trained to preserve the input image unchanged when the identity embedding is fused into the SDTB through SDCA.

**Ratio Control Embedding.** OneRestore is designed to fully restore a clean image regardless of the severity of degradation, as long as a corresponding degradation embedding is provided. This observation suggests the potential for controlling the degree of restoration by adjusting the degradation embedding vector. To enable this, we leverage an identity embedding that preserves the input without any modification. Specifically, we linearly combine the identity embedding and the degradation embedding $\mathbf{e}_d$ according to the desired restoration intensity ratio, as follows:

$$\mathbf{e}_r = (1 - w) \cdot \mathbf{e}_i + w \cdot \mathbf{e}_d, \tag{2}$$

where $w$ is an intensity ratio control parameter ranging from identity ($w = 0$) to full restoration ($w = 1$), and $\mathbf{e}_r$ denotes the corresponding ratio control embedding. Adjusting $w$ enables the model to continuously control the restoration intensity for each degradation type, allowing degradations to be removed partially rather than entirely. Note that the ratio control embedding, like the identity and other degradation embeddings, is integrated into the SDTB of the baseline restoration model via SDCA, and guides the model to remove degradations proportionally to the specified ratio.

## 3.3 LOSS FUNCTIONS

Based on the identity embedding and ratio control embedding, we perform disentangled learning guided by the following four loss functions.

**Identity Loss.** The identity loss is designed to ensure that the model preserves the original content when no degradation needs to be removed. In other words, it enforces the model to maintain the input unchanged when the identity embedding is used as a prompt in the restoration model, as follows:

$$\mathcal{L}_{\text{iden}} = \mathcal{L}_{\text{L1-SSIM}}(I, \hat{I}), \tag{3}$$

where $I$ is the input image, $\hat{I}$ represents the model output with identity embedding. Through this loss term, the model learns to behave as an identity function when necessary, which can be utilized for the intermediate and permutation-invariant losses in cases involving a single degradation type.

**Ratio Control Loss.** The goal of the ratio control loss is to ensure that, when the ratio control embedding is used as a prompt, the model removes degradations proportionally to the specified ratio, rather than performing full restoration. Ideally, this would require GT images corresponding to each value of $w$ in equation 2, where only the $w$ proportion of the degradation is removed. However, in practice, constructing GT images for every possible $w$ is infeasible. Therefore, during training, we only generate and use GT images corresponding to $w = 0.5$. In the experiments, we demonstrate that training with only the $w = 0.5$ case generalizes well, and the model performs seamlessly for other $w$ values during inference. Accordingly, the ratio control loss is defined using only the half-ratio case during training, as follows:

$$\mathcal{L}_{\text{ratio}} = \mathcal{L}_{\text{L1-SSIM}}(J_h, \hat{J}_h) + \mathcal{L}_{\text{L1-SSIM}}(J, \hat{J}_{hh}), \tag{4}$$

where $J_h$ denotes the half-degraded GT image, $\hat{J}_h$ is the corresponding half-restored output, $J$ is the fully clean GT image, and $\hat{J}_{hh}$ is the fully restored output obtained by reapplying the restoration model to $\hat{J}_h$. Note that both $\hat{J}_h$ and $\hat{J}_{hh}$ are generated using the ratio control embedding for $w = 0.5$. The first term supervises the half-restored output $\hat{J}_h$ to match the corresponding half-degraded GT $J_h$, enabling the restoration model to develop fine-grained controllability by learning to remove

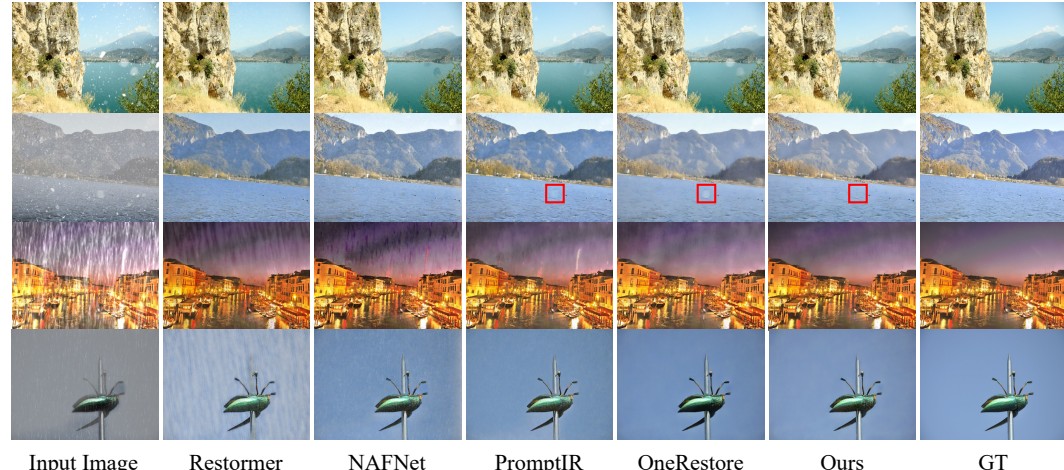

Input Image    Restormer    NAFNet    PromptIR    OneRestore    Ours    GT

Figure 2: Qualitative comparisons on the CCDD-11 dataset. From top to bottom, each row corresponds to a case in which the input images are degraded by $snow$, $low + haze + snow$, $rain$, and $low + haze + rain$, respectively.

approximately half of the degradation, as intended by the half-intensity ratio control embedding. The second term applies the same half-intensity ratio control embedding, encouraging the resulting output $\hat{J}_{\text{hh}}$ to progressively approach the fully restored clean image $J$. This enforces a self-consistency constraint that reinforces the linearity of the control space.

**Intermediate Loss.** To improve controllability and address the challenges of sequential restoration under multiple degradations, we propose an intermediate loss function that provides precise and explicit supervision for removing each specific degradation at its respective stage in the restoration process. Let $D_A$ and $D_B$ denote two independent degradations present in a composite image. If the image is restored sequentially by first removing $D_A$ and then $D_B$, or in reverse order, any errors introduced in the first step are likely to propagate into the second. To mitigate this issue, the intermediate loss is designed to help the model handle each degradation independently during the restoration process, as follows:

$$\mathcal{L}_{\text{inter}} = \mathcal{L}_{\text{L1-SSIM}}(J_{\text{A}}, \hat{J}_{\text{A}}) + \mathcal{L}_{\text{L1-SSIM}}(J_{\text{B}}, \hat{J}_{\text{B}}), \tag{5}$$

where $J_{\text{A}}$ denotes the GT image with degradation $D_A$ removed (but not completely clean), and $\hat{J}_{\text{A}}$ is the model output when restoring only $D_A$. Similarly, $J_{\text{B}}$ is the GT image with degradation $D_B$ removed, and $\hat{J}_{\text{B}}$ is the corresponding output when the model restores only $D_B$. Note that equation 5 can be easily extended to cases involving more than two degradations. Also, for single-degradation scenarios, one of $D_A$ or $D_B$ can be treated as "clear," and the identity embedding can be applied accordingly.

**Permutation-Invariant Loss.** To ensure consistency in sequential restoration of composite degradations, we propose a permutation-invariant loss that enforces the final output to remain unaffected by the order in which degradations are removed. To ensure consistent results regardless of the restoration order, our permutation-invariant loss consists of two terms as follows:

$$\mathcal{L}_{\text{per1}} = \mathcal{L}_{\text{L1-SSIM}}(J, \hat{J_{\text{AB}}}) + \mathcal{L}_{\text{L1-SSIM}}(J, \hat{J_{\text{BA}}}), \quad \mathcal{L}_{\text{per2}} = \mathcal{L}_{L1}(\hat{J_{\text{AB}}}, \hat{J_{\text{BA}}}), \tag{6}$$

where $J$ represents the GT image, and $\hat{J}_{\text{AB}}$ represents the output when the model restores the image in the order of $D_A$ followed by $D_B$. Similarly, $\hat{J}_{\text{BA}}$ represents the model output when the restoration order is reversed. The $\mathcal{L}_{\text{per1}}$ ensures that both restoration sequences produce outputs close to the GT image $J$. The smooth L1 loss $\mathcal{L}_{\text{per2}}$ enforces consistency between the two restoration outputs, $\hat{J}_{\text{AB}}$ and $\hat{J}_{\text{BA}}$.

Finally, the restoration model is trained using the baseline loss equation 1, along with the four proposed loss functions: identity equation 3, ratio control equation 4, intermediate equation 5, and permutation-invariant loss functions equation 6.

Table 1: Quantitative comparisons on CCDD-11 dataset[1].

| Types | Methods | PSNR ↑ | SSIM ↑ | #Params |
|---|---|---|---|---|
| | Input | 16.04 | 0.61 | - |
| One-to-One | MIRNet (Zamir et al., 2020) | 26.17 | 0.87 | 31.79M |
| | MPRNet (Zamir et al., 2021a) | 27.07 | 0.86 | 15.74M |
| | MIRNetV2 (Zamir et al., 2022b) | 24.90 | 0.83 | 15.74M |
| | Restormer (Zamir et al., 2022a) | 25.43 | 0.84 | 26.13M |
| | DGUNet (Mou et al., 2022) | 27.17 | 0.86 | 17.33M |
| | NAFNet (Chen et al., 2022a) | 26.78 | 0.80 | 17.11M |
| | Fourmer (Zhou et al., 2023a) | 24.49 | 0.82 | 0.55M |
| | OKNet (Cui et al., 2024a) | 27.54 | 0.87 | 4.72M |
| One-to-Many | AirNet (Li et al., 2022) | 26.33 | 0.84 | 8.93M |
| | AdaIR (Cui et al., 2024b) | 27.25 | 0.86 | 28.77M |
| | PromptIR (Potlapalli et al., 2023) | 28.07 | 0.87 | 38.45M |
| | WGWSNet-WG (Zhu et al., 2023) | 25.40 | 0.85 | 25.76M |
| | WGWSNet-WS (Zhu et al., 2023) | 20.20 | 0.74 | 25.76M |
| One-to-Composite | OneRestore (Visual Embedding) (Guo et al., 2024) | 27.38 | 0.86 | 5.98M |
| | OneRestore (Text Embedding) (Guo et al., 2024) | 27.74 | 0.87 | 5.98M |
| | Ours (Visual Embedding) | 27.89 | 0.86 | 5.98M |
| | Ours (Text Embedding) | **28.28** | 0.87 | 5.98M |

Table 2: Ablation study on different loss functions.

| Identity | Ratio control | Intermediate | Permutation-invariant | PSNR |
|---|---|---|---|---|
| ✓ | | | | 27.85 |
| ✓ | ✓ | | | 27.89 |
| ✓ | | ✓ | | 28.13 |
| ✓ | | | ✓ | 28.20 |
| ✓ | ✓ | ✓ | ✓ | **28.28** |

Table 3: PSNR evaluation under the identity embedding. The *clear* prompt is used for OneRestore.

| Method | Low | Haze | Rain | Snow | Low+Haze | Low+Rain |
|---|---|---|---|---|---|---|
| Guo et al. (2024) | 21.31 | 23.87 | 25.04 | 24.94 | 24.45 | 23.89 |
| Ours | 55.27 | 57.31 | 57.35 | 57.44 | 57.58 | 57.24 |

| Method | Low+Snow | Haze+Rain | Haze+Snow | Low+Haze+Rain | Low+Haze+Snow | Average |
|---|---|---|---|---|---|---|
| Guo et al. (2024) | 23.34 | 23.60 | 23.80 | 23.72 | 23.67 | 23.78 |
| Ours | 57.00 | 57.32 | 57.59 | 57.66 | 57.73 | **57.05** |

## 4 EXPERIMENTS

### 4.1 EXPERIMENT SETTINGS

**Implementation Details.** Our DisIR uses PyTorch 1.12.0 and trains on 4 NVIDIA A100 GPUs, requiring approximately two days per stage. During the training phase of OneRestore (Guo et al., 2024), we use a learning rate of 1e-4. In subsequent stages, we start with a reduced learning rate of 2.5e-5, which is achieved by decreasing the initial rate by a factor of 1/4.

**Evaluation Dataset and Metrics.** We create Controllable Composite Degradation Dataset (CCDD-11), which follows the pipeline introduced in CDD-11 (Guo et al., 2024) (see Appendix section B for details) Specifically, our CCDD-11 adopts the same pipeline as CDD-11, while generating all pairs corresponding to individual degradation removal from composite degradation images. However, due to the limited variety of rain masks in the original CDD-11, we incorporate a more diverse range of rain masks into our CCDD-11. Restoration performance is evaluated using PSNR and SSIM two standard metrics.

**Compared Methods.** We compare our DisIR with seven one-to-one restoration models: MIRNet (Zamir et al., 2020), MPRNet (Zamir et al., 2021a), MIRNetV2 (Zamir et al., 2022b), Restormer (Zamir et al., 2022a), DGUNet (Mou et al., 2022), NAFNet (Chen et al., 2022a), and Fourmer (Zhou et al., 2023a); four one-to-many models: AirNet (Li et al., 2022), AdaIR (Cui et al., 2024b), PromptIR (Pot-

---

[1]The performance gap compared to the originally reported OneRestore (Guo et al., 2024) results is primarily due to modifications made to all rain-related datasets. Unlike the original setup, which uses a limited variety of rain masks in the CDD-11 dataset, our version incorporates a broader and more diverse set of rain mask variations.

Table 4: Embedder classification accuracy according to changes in the intensity ratio.

| Intensity Ratio $w$ | 1.0 | 0.9 | 0.8 | 0.7 | 0.6 | 0.5 | 0.4 | 0.3 | 0.2 | 0.1 | 0.0 |
|---|---|---|---|---|---|---|---|---|---|---|---|
| Accuracy | 0.3 % | 0.4 % | 0.8 % | 2.5 % | 16.4 % | 43.8 % | 56.2 % | 77.9 % | 94.7 % | 95.0 % | 94.9 % |

Table 5: PSNR evaluation of selective restoration on images with two combined degradations.

| Degradation | Task | Guo et al. (2024) | Ours | Task | Guo et al. (2024) | Ours |
|---|---|---|---|---|---|---|
| Low+Haze | Delow | 20.89 | 30.61 | Dehaze | 22.56 | 35.50 |
| Low+Rain | Delow | 24.67 | 26.74 | Derain | 28.76 | 36.37 |
| Low+Snow | Delow | 23.42 | 26.68 | Desnow | 27.53 | 34.30 |
| Haze+Rain | Dehaze | 27.73 | 33.01 | Derain | 34.69 | 40.30 |
| Haze+Snow | Dehaze | 25.56 | 33.13 | Desnow | 31.61 | 37.81 |
| Average | - | 24.45 | **30.03** | - | 29.03 | **36.86** |

Table 6: PSNR evaluation of selective restoration on images with three combined degradations.

| Degradation | Double Restoration | | Single Restoration | |
|---|---|---|---|---|
| | Guo et al. (2024) | Ours | Guo et al. (2024) | Ours |
| Low+Haze+Rain | 22.33 | 27.00 | 25.10 | 32.50 |
| Low+Haze+Snow | 21.29 | 25.42 | 24.27 | 31.48 |
| Average | 21.81 | **26.21** | 24.69 | **31.99** |

lapalli et al., 2023), and WGWS (Zhu et al., 2023); and one one-to-composite model, OneRestore (Guo et al., 2024). All models are trained on the CCDD-11 training set. During our disentangled learning, we exclude images containing three types of degradation due to the impractical number of possible combinations. However, we include these samples in the test set to evaluate the generalization ability of our DisIR. For PromptIR, since the original paper does not specify the optimal number of learnable prompts, we train the model using both five and eleven prompts. The version with five prompts achieves better performance and is thus reported. For WGWS, we set weather-specific parameters to cover all degradation types in CCDD-11. WG refers to the first stage, and WS refers to the second.

## 4.2 COMPARISONS

As reported in Tab. 1, our DisIR achieves the highest performance when using text-based degradation embeddings. Even when using visual-based embeddings extracted directly from input images, our DisIR generally outperforms existing approaches. Interestingly, although not originally designed as a One-to-Composite model, PromptIR outperforms OneRestore on the CCDD-11 dataset. However, it is worth noting that PromptIR relies on a much larger model (38.45M parameters vs. 5.98M) and, by design, is not easily extended for controllability. As shown in Fig. 2, our DisIR more effectively removes degradations compared to existing approaches in terms of visual quality. Especially, in the second row of Fig. 2, the red box highlights a challenging case where both PromptIR and OneRestore leave noticeable snow artifacts, whereas our DisIR achieves more effective removal under composite degradation ($low + haze + snow$). We believe our disentangled learning approach provides more effective guidance, offering not only enhanced controllability but also superior performance in composite degradation restoration.

## 4.3 ABLATION STUDIES

In this subsection, we conduct ablation studies to evaluate the effectiveness of the proposed loss functions, identity embedding, and ratio control embedding. Detailed settings for each ablation experiment and additional qualitative results are provided in the supplementary material.

**Effect of Losses.** To verify the effectiveness of the proposed loss functions, we conduct an ablation study by excluding each component on the CCDD-11 dataset, as shown in Tab. 2. While identity loss alone leads to improved performance compared to OneRestore, incorporating additional loss functions results in further performance gains. The best performance is achieved when all loss components are used together, indicating that the proposed loss functions effectively guide the one-to-composite model to disentangle mixed degradations.

**Effect of Identity Embedding.** The identity embedding is designed to guide the restoration model to preserve the input image, even in the presence of degradation. To verify its effectiveness, we measure

Table 7: Classification performance after selective restoration. The top-row degradation types represent the remaining degradations after selective restoration.

| Method | Low | Haze | Rain | Snow | Low+Haze | Low+Rain | Low+Snow | Haze+Rain | Haze+Snow | Average |
|---|---|---|---|---|---|---|---|---|---|---|
| Guo et al. (2024) | 76.7% | 61.7% | 73.8% | 48.6% | 76.7% | 42.0% | 24.5% | 60.5% | 15.5% | 53.33% |
| Ours | 98.6% | 85.5% | 93.7% | 94.3% | 88.7% | 91.6% | 98.4% | 95.6% | 94.6% | **93.44%** |

Table 8: PSNR evaluation with varying restoration orders.

| Degradation | Task | Guo et al. (2024) | Ours | Task | Guo et al. (2024) | Ours | Task | Guo et al. (2024) | Ours |
|---|---|---|---|---|---|---|---|---|---|
| Low+Haze | Delow → Dehaze | 20.95 | 25.80 | Dehaze → Delow | 21.20 | 24.76 | Dehaze + Delow | 25.07 | 25.28 |
| Low+Rain | Delow → Derain | 22.66 | 25.77 | Derain → Delow | 24.45 | 25.54 | Derain + Delow | 25.27 | 25.41 |
| Low+Snow | Delow → Desnow | 21.63 | 25.26 | Desnow → Delow | 22.63 | 24.87 | Desnow + Delow | 24.64 | 24.95 |
| Haze+Rain | Dehaze → Derain | 27.74 | 30.12 | Derain → Dehaze | 28.92 | 30.75 | Derain + Dehaze | 29.62 | 30.52 |
| Haze+Snow | Dehaze → Desnow | 26.26 | 28.66 | Desnow → Dehaze | 27.13 | 29.59 | Desnow + Dehaze | 28.65 | 29.42 |
| Average | - | 23.85 | **27.12** | - | 24.87 | **27.10** | - | 26.65 | **27.12** |

Table 9: PSNR evaluation on the Blur-Noise-JPEG composite degradation scenario.

| Method | Blur | Noise | JPEG | Blur+Noise | Blur+JPEG | Noise+JPEG | Blur+Noise+JPEG | Average |
|---|---|---|---|---|---|---|---|---|
| Guo et al. (2024) | 28.04 | 31.94 | 30.47 | 25.17 | 25.04 | 28.80 | 24.48 | **27.70** |
| Ours | 30.40 | 32.45 | 30.93 | 25.34 | 25.14 | 28.85 | 24.60 | **28.24** |

the PSNR between the input and the output generated using the identity embedding. OneRestore For comparison, we evaluate OneRestore with the $clear$ prompt, which most closely resembles the identity condition in its framework. As shown in Tab. 3, our DisIR yields outputs with minimal deviation from the inputs, demonstrating superior identity preservation compared to OneRestore and validating the effectiveness of our identity embedding.

**Effect of Ratio Control Embedding.** The ratio control embedding is designed to guide the intensity level of restoration, such that only a specified proportion of the degradation is removed. To achieve this, we construct a prompt by linearly combining the identity embedding and the target degradation embedding according to equation 2. As shown in Fig. 3, this allows for successful control over the degree of restoration. To quantitatively evaluate controllability, we measure classification accuracy using the ratio control embedding generated with varying values of $w$ in equation 2. Note that $w = 1$ leads to full degradation removal and low classifier accuracy, while $w = 0$ preserves the degradation and yields high accuracy. As reported in Tab. 4, the classification performance varies with $w$, indicating that our DisIR effectively enables intensity-level control over restoration.

**Selective Restoration.** Thanks to our disentangled learning pipeline, the proposed model excels at selectively removing a target degradation from composite images while preserving other degradations, outperforming OneRestore in this setting. For instance, in an image degraded by both $haze$ and $snow$, selective restoration involves removing only one component (*e.g.*, $haze$), resulting in an output that retains the other (*e.g.*, $snow$). In Tab. 5, we report the performance of our model on selective restoration tasks involving two degradations. Specifically, for each image containing two degradation types, we remove only one and evaluate the output against a GT image in which only the target degradation has been removed. For example, in the case of a composite degradation of $low + haze$, the GT for the $delow$ task is a clean image synthesized with only $haze$. Although OneRestore can also apply a single degradation embedding, as shown in Tab. 5, our model consistently achieves higher performance across all selective restoration tasks. We further extend this evaluation to images containing three degradation types. In this setting, we test restoration by removing either one or two degradations and compare the results with the corresponding GT images. Due to the large number of possible restoration combinations, we report the average performance across all single- and double-degradation removal tasks from triple-composite inputs. As reported in Tab. 6, our model again outperforms OneRestore in all settings. To further assess whether our DisIR effectively disentangles multiple degradations, we conduct an analysis using a classification model. Specifically, for images containing multiple degradations, we perform selective restoration to remove only the target degradation and then examine whether the remaining degradations are preserved in the output. This is done by extracting degradation embeddings from the selectively restored images and performing classification to identify the remaining degradation types. As shown in Tab. 7, our DisIR consistently outperforms OneRestore in identifying the remaining degradations, regardless of the type of degradation being removed. Moreover, as shown in Fig. 3, our qualitative results demonstrate that the selective restoration closely matches the corresponding GT images. These results demonstrate

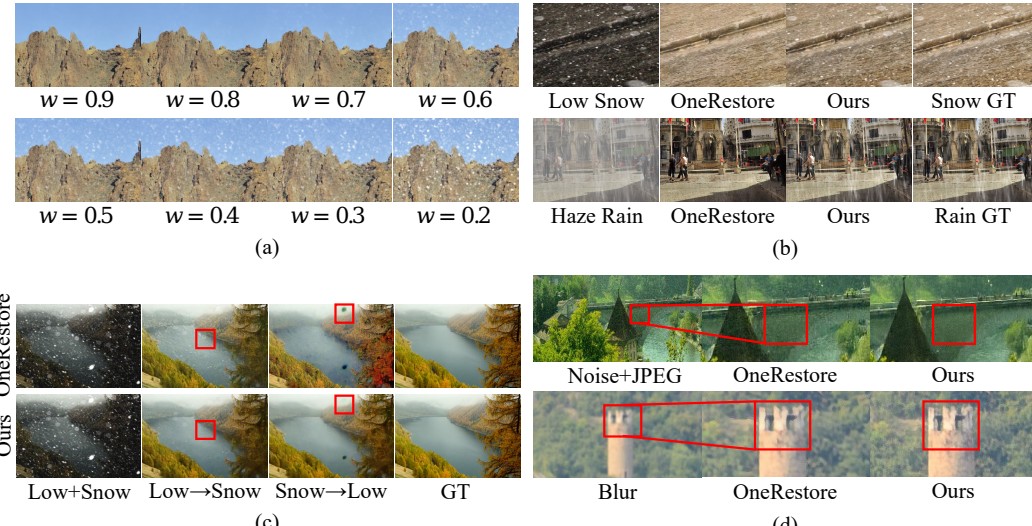

Figure 3: (a) Ratio control embedding. (b) Selective restoration. (c) Restoration order dependency. (d) Blur-noise-JPEG composite degradation.

that our model enables fine-grained controllability by providing disentangling guidance, allowing it to decompose complex composite degradations into their constituent components.

**Restoration Order Dependency.** Under composite degradations, our model is capable of sequentially removing each degradation individually. As shown in Tab. 8, the model trained with our disentangled learning pipeline consistently achieves strong performance regardless of the removal order, in contrast to OneRestore. This order-invariant behavior is attributed to the permutation-invariant loss equation 6, and demonstrates the model's ability to effectively disentangle and target individual degradations in inputs affected by multiple degradation types. Notably, even when performing restoration in a single step using the composite embedding vectors, our DisIR still outperforms OneRestore, further validating its robustness and controllability. This can also be visually confirmed in Fig. 3.

**Blur-Noise-JPEG Composite Degradation.** To further validate the generalizability of the proposed disentangled learning pipeline, we conduct experiments not only on diverse weather removal but also on a composite degradation scenario involving $blur + noise + JPEG$. As reported in Tab. 9, our DisIR consistently outperforms OneRestore across all cases. Experiments in more diverse settings, including real-world data, are provided in the supplementary material.

## 5 CONCLUSION AND LIMITATIONS

In this paper, we proposed a disentangled learning pipeline for controllable all-in-one image restoration under composite degradation scenarios. Our DisIR uses four objectives: (1) identity loss for preserving input when no degradation is present, (2) ratio control loss for adjusting restoration intensity, (3) intermediate loss for selective degradation removal, and (4) permutation-invariant loss ensuring order-consistent results. These components collectively enable the model to effectively disentangle and selectively restore target degradations without affecting unrelated image content. Our method supports fine-grained intensity control and achieves state-of-the-art performance on composite degradation benchmarks. Our findings underscore the importance of disentangled representations and targeted guidance in building flexible and controllable restoration systems for complex real-world scenarios. However, our approach has several limitations. First, the model may struggle to generalize to unknown degradations not encountered during training, indicating a need to extend the framework toward handling unseen or open-set degradation types. Second, while effective for up to three concurrent degradations, the performance may degrade when more types are combined, especially in the presence of spatially varying degradations. Addressing these challenges would further enhance the robustness and scalability of controllable restoration systems in real-world applications.

## ETHICS STATEMENT

Our research aims to restore images degraded by adverse weather conditions, with the goal of positive societal applications such as improving the safety of autonomous systems. The datasets used for training and evaluation, including our proposed CCDD-11, were constructed using publicly available academic benchmarks that do not contain sensitive or personal information. We do not foresee any direct negative societal consequences or ethical concerns arising from this work, as its scope is limited to degradation removal rather than applications that might compromise privacy or security.

## REPRODUCIBILITY STATEMENT

To ensure the reproducibility of our results, we provide the following details. The implementation of our proposed DisIR framework, including the architecture and the four novel loss functions (identity, ratio control, intermediate, and permutation-invariant), is described in section 3. Our newly created CCDD-11 is detailed in Appendix section B, which explains the generation pipeline and the inclusion of diverse rain masks. All implementation details, such as hyperparameters, training procedures on NVIDIA A100 GPUs, and evaluation settings, are provided in section 4.1 and Appendix section C. The complete source code, training scripts, and the generation code for our CCDD-11 dataset will be made publicly available upon publication.

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

## A  SUMMARY OF APPENDIX

In this appendix, we first describe the construction of our dataset for selective restoration and half-restoration (see Section B), and provide detailed explanations of our training settings and configuration details (see Section C). We then present comprehensive descriptions of experimental settings for the main experiments in Section D. Experimental results that demonstrate controllability on the Blur–Noise–JPEG Composite Dataset are provided in Section E, and results on real-world datasets are presented in Section F. In Section G, we present qualitative evaluation results for various scenarios, including identity operation, ratio control restoration, selective restoration, restoration order dependence, and composite image restoration. Finally, we provide a detailed account of Large Language Model usage in our research and writing process in Section H.

## B  CONTROLLABLE COMPOSITE DEGRADATION DATASET FORMULATION

### B.1  DATASET CONSTRUCTION NECESSITY

Our work addresses controllable restoration tasks that existing datasets cannot support. Specifically, DisIR requires ground truth pairs that are absent in current public datasets: (1) variable-intensity degradation images with identical content and context but varying degradation intensities for ratio control training, and (2) selective restoration pairs where only specific degradations are removed from composite images while preserving others. Without these pairs, it is impossible to train or evaluate models on fine-grained controllability tasks. Therefore, the Controllable Composite Degradation Dataset (CCDD-11) was created not as an optional addition, but as an essential foundation for controllable composite image restoration research. To ensure objectivity, CCDD-11 strictly follows the original CDD-11 synthesis methodology with transparent, reproducible extensions using fixed random seeds and uniform parameter distributions.

### B.2  CCDD-11 DATA PIPELINE

To construct CCDD-11, we generally follow the protocol of the CDD-11 (Guo et al., 2024) dataset, with several modifications to enable controllable restoration. Specifically, we introduce selective restoration pairs and half-restoration pairs, which allow for more flexible and targeted restoration scenarios. To further enrich the dataset, we incorporate a wider variety of rain masks, which increases the diversity of degradation patterns in CCDD-11. For the source images, we select 1,383 high-resolution clean images from the RAISE (Dang-Nguyen et al., 2015) dataset and uniformly resize them to a resolution of 1080×720 pixels. Of these, 1,183 images are used for training and the remaining 200 for testing. The procedure for generating composite degradation images is as follows:

$$I(x) = \mathcal{D}_h(\mathcal{D}_{rs}(\mathcal{D}_l(J(x)))), \tag{7}$$

where $I(x)$ denotes the degraded image, $J(x)$ is the corresponding clean image as described above, $\mathcal{D}_h$ represents haze degradation, $\mathcal{D}_{rs}$ represents rain or snow degradation, and $\mathcal{D}_l$ represents low-light degradation. Each of $\mathcal{D}_h$, $\mathcal{D}_{rs}$, and $\mathcal{D}_l$ can be applied independently or omitted during the degradation process. The order and combination of these degradations follow the protocol established in CDD-11. Note that in addition to the fully composite degraded images generated by Eq. equation 7, we also store images degraded by only a subset of these degradations. For these subset images, we use exactly the same degradation masks and parameters as in the composite degradations, ensuring that they can serve as ground truth for selective restoration tasks. For example, when generating an image with $haze + rain$ degradations, we also store the corresponding $haze$ and $rain$ images, which are produced using the same degradation masks and parameters, following the same degradation pipeline. We also define half-degradation images as those generated by applying the same degradation process but with approximately half the intensity of the original parameters. These half-degradation images are also stored alongside the fully degraded images to enable our proposed intensity ratio control restoration.

**Low-Light.** According to Retinex theory, the low-light image generation pipeline is defined as follows:

$$I_l(x) = \mathcal{D}_l(x) = \frac{J(x)}{L(x)} \cdot L(x)^\gamma + \varepsilon, \tag{8}$$

where $J(x)$ denotes the clean image and $I_l(x)$ is the corresponding low-light image. In this formulation, $\gamma$ is a darkening coefficient ranging from 2 to 3, which serves as a brightness adjustment factor for the illumination map $L(x)$ generated by LIME (Guo et al., 2016). Gaussian noise $\varepsilon$ with zero mean and variance between 0.03 and 0.08 is added to better simulate low-light environments.

$$I_{l_h}(x) = \frac{J(x)}{L(x)} \cdot L(x)^{\gamma/2} + \varepsilon, \tag{9}$$

where $I_{l_h}(x)$ denotes the half-low-light image, which is generated by applying the low-light pipeline with the $\gamma$ parameter set to half the value used in Eq. equation 8.

**Rain/Snow Streaks.** Following (Chen et al., 2021b), we add snow streaks to images using alpha blending. Similarly, our rain synthesis, based on (Li et al., 2019), is modified to use alpha blending, enabling ratio control through adjustable weighting. The pipeline for synthesizing rain and snow streaks is as follows:

$$I_{rs}(x) = D_{rs}(D_l(x)) = I_l(x)(1 - \mathcal{RS}) + \mathcal{RS}, \tag{10}$$

where $I_{rs}(x)$ denotes the rain/snow streak image, and $\mathcal{RS}$ represents rain mask or snow mask. The rain mask is sourced from (Garg & Nayar, 2006), while the snow mask is sourced from (Liu et al., 2018).

$$I_{rs_h}(x) = I_l(x)(1 - 0.5 \cdot \mathcal{RS}) + 0.5 \cdot \mathcal{RS}, \tag{11}$$

where $I_{rs_h}(x)$ denotes the half-intensity rain/snow streak image, generated by applying alpha blending with the weight for rain or snow reduced to half of that used in Eq. equation 10.

**Haze.** Haze degradation is introduced into our pipeline using the atmospheric scattering model, as follows:

$$t = e^{-\beta \cdot d(x)}, \quad t_h = e^{-(\beta/2) \cdot d(x)}, \tag{12}$$

where $t$ is the transmission map, $\beta$ is the haze density coefficient, and $d(x)$ is the depth information estimated from MegaDepth (Li & Snavely, 2018). Here, $t_h$ represents the half-transmission map, which is obtained by setting the haze density coefficient to $\beta/2$. The value of $\beta$, which controls the haze density, is set in the range [1.0, 2.0].

$$I_h(x) = D_h(D_{rs}(D_l(x))) = I_{rs}(x)t + A(1 - t), \tag{13}$$

where $I_h(x)$ is the haze-degraded image, and $A$ is the atmospheric light, which is constrained to the range [0.6, 0.9].

$$I_{h_h}(x) = I_{rs}(x)t_h + A(1 - t_h), \tag{14}$$

where $I_{h_h}(x)$ is the half-haze image, in which the haze density is approximately half that of $I_h(x)$ due to the use of $t_h$.

### B.3 CROSS-DATASET CLASSIFICATION EVALUATION

We examine the generalization capabilities of models trained on CCDD-11 versus CDD-11 through comprehensive cross-evaluation experiments. During our analysis, we discovered that the original CDD-11 uses a limited and repetitive set of rain masks, while CCDD-11 addresses this limitation by incorporating 4,723 unique rain masks, significantly increasing diversity. This fundamental difference in dataset construction is reflected in our cross-evaluation results shown in Tab. 10. Models trained on CCDD-11 maintain robust performance when evaluated on CDD-11, with accuracy dropping only from 92% to 88%. In contrast, models trained on CDD-11 show performance degradation when evaluated on CCDD-11, with accuracy dropping from 98% to 59%. This asymmetric generalization pattern demonstrates that models trained on the limited and repetitive degradation patterns of CDD-11 fail to generalize to more diverse scenarios, whereas CCDD-11 yields models that are robust to both simple and complex degradations. These results validate CCDD-11 as a more comprehensive and challenging benchmark, confirming that its increased diversity provides superior training compared to the restricted variations in CDD-11.

### B.4 BLUR-NOISE-JPEG

To demonstrate the generalizability of our proposed method, we constructed not only the weather-related CCDD-11 dataset described above, but also an additional dataset incorporating camera-related

Table 10: Rain-related degradation classification accuracy across different training and testing scenarios.

| Degradation Type | (1) Trained on CCDD-11 Test on CCDD-11 | (2) Trained on CCDD-11 Test on CDD-11 | (3) Trained on CDD-11 Test on CCDD-11 | (4) Trained on CDD-11 Test on CDD-11 |
|---|---|---|---|---|
| rain | 98 % | 96 % | 75 % | 99 % |
| low rain | 99 % | 86 % | 58 % | 97 % |
| haze rain | 95 % | 90 % | 58 % | 98 % |
| low haze rain | 76 % | 82 % | 46 % | 95 % |
| **Average** | **92 %** | **88 %** | **59 %** | **98 %** |

and environment-related degradations such as blur, noise, and JPEG compression. In constructing the Blur-Noise-JPEG dataset, degradations are applied in the order of blur, noise, and JPEG compression. This sequence reflects the typical image formation process, in which optical blur occurs first, followed by sensor noise and then compression, as described in recent works such as Real-ESRGAN (Wang et al., 2021).

$$I(x) = \mathcal{D}_J(\mathcal{D}_n(\mathcal{D}_b(J(x)))), \tag{15}$$

where $I(x)$ denotes the final degraded image, $J(x)$ is the corresponding clean image, $\mathcal{D}_b$ represents blur degradation, $\mathcal{D}_n$ denotes the addition of Gaussian noise, and $\mathcal{D}_J$ indicates JPEG compression. The degradations are applied sequentially in the order of blur, noise, and JPEG compression, resulting in seven types of degradation in the dataset: Blur, Noise, JPEG, Blur+Noise, Blur+JPEG, Noise+JPEG, and Blur+Noise+JPEG. Similar to CCDD-11, the Blur-Noise-JPEG dataset also stores both subset degradation images and half-degradation images for each sample.

**Blur.** Blur degradation simulates the loss of sharpness that typically results from camera defocus or motion. The process for generating blurred images is defined as follows:

$$I_b(x) = J(x) * G(x; k, \sigma_b), \qquad I_{b_h}(x) = J(x) * G(x; k, \sigma_b/2), \tag{16}$$

where $I_b(x)$ and $I_{b_h}(x)$ denote blurred and half-blurred images respectively, $J(x)$ is the clean image, $*$ represents the convolution operation, and $G(x; k, \sigma_b)$ is a 2D Gaussian kernel with kernel size $k$ and standard deviation $\sigma_b$. The half-blurred image is generated using a standard deviation of $\sigma_b/2$.

**Noise.** Noise degradation simulates random pixel fluctuations resulting from sensor imperfections during image acquisition, and is defined as follows:

$$I_n(x) = J(x) + \mathcal{N}(0, \sigma_n^2), \qquad I_{n_h}(x) = J(x) + \mathcal{N}(0, (\sigma_n/2)^2), \tag{17}$$

where $I_n(x)$ and $I_{n_h}(x)$ denote noisy and half-noisy images, respectively. In this equation, $J(x)$ is the clean image, and $\mathcal{N}(0, \sigma_n^2)$ denotes zero-mean Gaussian noise with standard deviation $\sigma_n$. The half-noisy image is generated using a standard deviation of $\sigma_n/2$.

**JPEG.** JPEG degradation simulates compression artifacts that are typically introduced during image encoding and storage. The process for generating JPEG-compressed images is defined as follows:

$$I_j(x) = \text{JPEG}(J(x); q), \qquad I_{j_h}(x) = \text{JPEG}(J(x); q/2), \tag{18}$$

where $I_j(x)$ and $I_{j_h}(x)$ denote the JPEG-compressed and half-JPEG images, respectively. In this equation, $J(x)$ is the clean image, and $\text{JPEG}(J(x); q)$ denotes the operation of compressing $J(x)$ using the JPEG algorithm with quality factor $q$. The half-JPEG image is generated using a quality factor of $q/2$.

## C   TRAINING DETAILS

We first train the baseline model until its performance saturates, for up to 300 epochs. Subsequently, training is resumed from the saturated checkpoint with the learning rate reduced to $2.5 \times 10^{-5}$, using the Adam optimizer and incorporating our proposed novel loss functions. During this phase, the model is trained with a batch size of 6, and the learning rate is halved every 30 epochs. After approximately two days of training (about 60 epochs), the model typically achieves a PSNR of 28.16 on the main composite image restoration task. To ensure the model fully saturates and achieves its best possible performance, we further extend training to 200 epochs, where the PSNR saturates at 28.28. All weight hyperparameters from the baseline are kept unchanged at $(0.6, 0.3, 0.1)$ throughout the training process. For training, we use 1,183 images, each with 11 different degradation types and corresponding clear images, resulting in a total of 13,013 training samples. For testing, the model is evaluated on 200 images, each with 11 degradation types.

Table 11: Embedder classification accuracy according to changes in the intensity ratio.

| Intensity Ratio $w$ | 1.0 | 0.9 | 0.8 | 0.7 | 0.6 | 0.5 | 0.4 | 0.3 | 0.2 | 0.1 | 0.0 |
|---|---|---|---|---|---|---|---|---|---|---|---|
| Accuracy | 1.5 % | 1.9 % | 4.1 % | 11.7 % | 16.4 % | 17.7 % | 18.8 % | 36.0 % | 54.4 % | 64.8 % | 63.8 % |

Table 12: PSNR evaluation of selective restoration on images with two combined degradations.

| Degradation | Task | Guo et al. (2024) | Ours | Task | Guo et al. (2024) | Ours |
|---|---|---|---|---|---|---|
| Blur+Noise | Deblur | 24.87 | 28.56 | Denoise | 36.21 | 37.99 |
| Blur+JPEG | Deblur | 24.71 | 26.92 | DeJPEG | 37.89 | 39.98 |
| Noise+JPEG | Denoise | 27.77 | 30.32 | DeJPEG | 24.74 | 25.02 |
| Average | - | 25.78 | **28.60** | - | 32.95 | **34.33** |

Table 13: PSNR evaluation of selective restoration on images with three combined degradations.

| Degradation | Double Restoration | | Single Restoration | |
|---|---|---|---|---|
| | Guo et al. (2024) | Ours | Guo et al. (2024) | Ours |
| Blur+Noise+JPEG | 27.22 | 28.09 | 27.36 | 29.65 |

## D  EXPERIMENTAL DETAILS

This section provides a detailed description of the experiments conducted on the CCDD-11 dataset, as presented in the main paper. The experimental procedure for Tab. 4 is as follows. For each degradation type, we vary the ratio control embedding from 0.0 to 1.0 and perform restoration accordingly. Each restored result is then classified using the Text/Visual Embedder, which can classify the remaining degradation types. The reported value is the average classification accuracy across all degradation types and intensity ratios. This setup is motivated by the fact that when $w = 0.0$ (i.e., identity), the image remains unchanged from the input degradation, and thus is expected to be classified with high accuracy as the corresponding degradation type. In contrast, when $w = 1.0$, the model restores the image to be close to a clean image, which should result in low classification accuracy. This approach is adopted because generating ground truth pairs for every possible value of $w$ is impractical. The experiments in Tab. 12 and Tab. 13 are designed to demonstrate that our DisIR can perform selective restoration, accurately removing only the targeted degradations while preserving the others. In Tab. 12, we evaluate the PSNR performance of restoring each individual degradation from composite images containing two types of degradation, using ground-truth pairs generated according to our dataset pipeline. Tab. 13 extends this experiment to composite images containing three types of degradation. In this setting, we consider two tasks: double restoration, which targets restoring two out of the three degradations (leaving one degradation in the image), and single restoration, which targets restoring only one degradation (leaving the other two degradations in the image). Since there are many possible combinations for each task, we report the average PSNR across all cases. The experiment in Tab. 14 is designed to demonstrate the mitigation of order dependency by comparing the performance of two-stage and one-stage restoration on composite images containing two types of degradation. For the composite degradation, Blur+Noise, the notations Deblur $\rightarrow$ Denoise and Denoise $\rightarrow$ Deblur represent two-stage restoration, where each degradation is removed sequentially in a different order. In contrast, Denoise + Deblur refers to the one-stage restoration, where both degradations are removed simultaneously in a single step.

## E  EXPERIMENTAL RESULTS ON BLUR–NOISE–JPEG COMPOSITE DATASET

The experiments summarized in Tab. 11, Tab. 12, Tab. 13, and Tab. 14 are also carried out on the Blur-Noise-JPEG dataset using the same experimental protocols as in the main paper. The overall reduction in accuracy observed in Tab. 11 likely results from the generally lower restoration performance on the Blur-Noise-JPEG dataset. However, the key observation is the linear change in accuracy as the ratio control parameter $w$ varies. This result demonstrates that the model is capable of controlling the restoration intensity, thereby demonstrating the contribution of our ratio control embedding and ratio control loss. As shown in Tab. 12 and Tab. 13, our DisIR achieves superior performance in selective restoration compared to previous approaches. The results in Tab. 14 indicate that our DisIR suppresses quality variations caused by order dependency.

Table 14: PSNR evaluation with varying restoration orders.

| Degradation | Task | Guo et al. (2024) | Ours | Task | Guo et al. (2024) | Ours | Task | Guo et al. (2024) | Ours |
|---|---|---|---|---|---|---|---|---|---|
| Blur+Noise | Deblur → Denoise | 21.97 | 26.01 | Denoise → Deblur | 23.95 | 25.99 | Deblur + Denoise | 25.24 | 25.33 |
| Blur+JPEG | Deblur → DeJPEG | 22.32 | 25.92 | DeJPEG → Deblur | 23.73 | 25.95 | Deblur + DeJPEG | 25.07 | 25.14 |
| Noise+JPEG | Denoise → DeJPEG | 27.02 | 29.41 | DeJPEG → Denoise | 26.94 | 29.42 | Denoise + DeJPEG | 28.80 | 28.84 |
| Average | - | 23.77 | **27.11** | - | 24.87 | **27.12** | - | 26.37 | **26.44** |

Table 15: NIQE scores on four Real-world datasets.

| NPE (Wang et al., 2013b) | | RS (Yang et al., 2017) | | Snow-100K-R (Liu et al., 2018) | | RTTS (Li et al., 2018) | |
|---|---|---|---|---|---|---|---|
| Method | NIQE ↓ | Method | NIQE ↓ | Method | NIQE ↓ | Method | NIQE ↓ |
| Liu et al. (2021) | 7.77 | Zamir et al. (2021b) | 3.55 | Liang et al. (2022) | 3.93 | Dong et al. (2020) | 4.77 |
| Ma et al. (2022) | **3.97** | Fu et al. (2021) | **3.27** | Chen et al. (2022c) | 3.13 | Guo et al. (2022) | 5.34 |
| Xu et al. (2022) | 4.49 | Wang et al. (2023a) | 3.32 | Kulkarni et al. (2022) | 3.34 | Zheng et al. (2023) | 5.03 |
| Guo et al. (2024) | 4.83 | Guo et al. (2024) | 3.67 | Guo et al. (2024) | 2.93 | Guo et al. (2024) | 4.76 |
| Ours | 4.44 | Ours | 3.66 | Ours | **2.92** | Ours | **4.53** |

# F    EXPERIMENTAL RESULTS ON REAL-WORLD DATASETS

We assess the robustness of our model trained on CDD-11 by conducting extensive real-world image restoration experiments under challenging degradation scenarios. To comprehensively address four distinct real-world degradations, we selected the following benchmarks: NPE (Wang et al., 2013b) for low-light enhancement, RTTS (Li et al., 2018) for dehazing, RS (Yang et al., 2017) for deraining, and Snow100k-R (Liu et al., 2018) for desnowing. Quantitative results are reported on these four real-world benchmarks using the no-reference quality metric Natural Image Quality Evaluator (NIQE). Tab. 15 shows that our model achieves competitive results in multiple datasets. Specifically, our DisIR outperforms previous approaches on RTTS and Snow100k-R, and achieves marginal improvements over OneRestore (Guo et al., 2024) on NPE and RS. The qualitative results in Fig. 4 also show that our DisIR produces better restoration quality than the previous approach.

# G    QUALITATIVE EVALUATION RESULTS

## G.1    IDENTITY OPERATION

To evaluate the identity operation, we present qualitative results showing that our model produces outputs identical to the inputs by employing the proposed identity embedding and identity loss. As shown in the qualitative results in Fig. 5 and Fig. 6, our DisIR successfully preserves the input without introducing unwanted alterations, as intended. For comparison, following the approach described in the main paper, we use the *clear* prompt in OneRestore (Guo et al., 2024), which serves as the closest equivalent to the identity condition in its framework. In our method, we employ the proposed identity embedding.

## G.2    RATIO CONTROL RESTORATION

With the proposed ratio control embedding and ratio control loss, our model is able to control the intensity of restoration. We present qualitative results that demonstrate ratio control across various types of degradation. By definition, $w = 0.0$ corresponds to the identity embedding and produces an output similar to the degraded input image, whereas $w = 1.0$ yields a fully restored image. As shown in Fig. 7, all composite degradations are effectively controlled in a linear fashion as $w$ varies. Fig. 8 presents qualitative ratio control restoration results for three composite degradation types that are excluded from ratio control loss training due to feasibility constraints, yet still demonstrate effective performance. Fig. 9 shows qualitative results for the same experiment conducted on the Blur-Noise-JPEG dataset. In these cases, as $w$ decreases, the outputs gradually approach the identity operation. Notably, our method demonstrates reliable linear control over the entire range of $w$ from 0 to 1, despite being trained only on half-degradation without explicit supervision for every possible value of $w$.

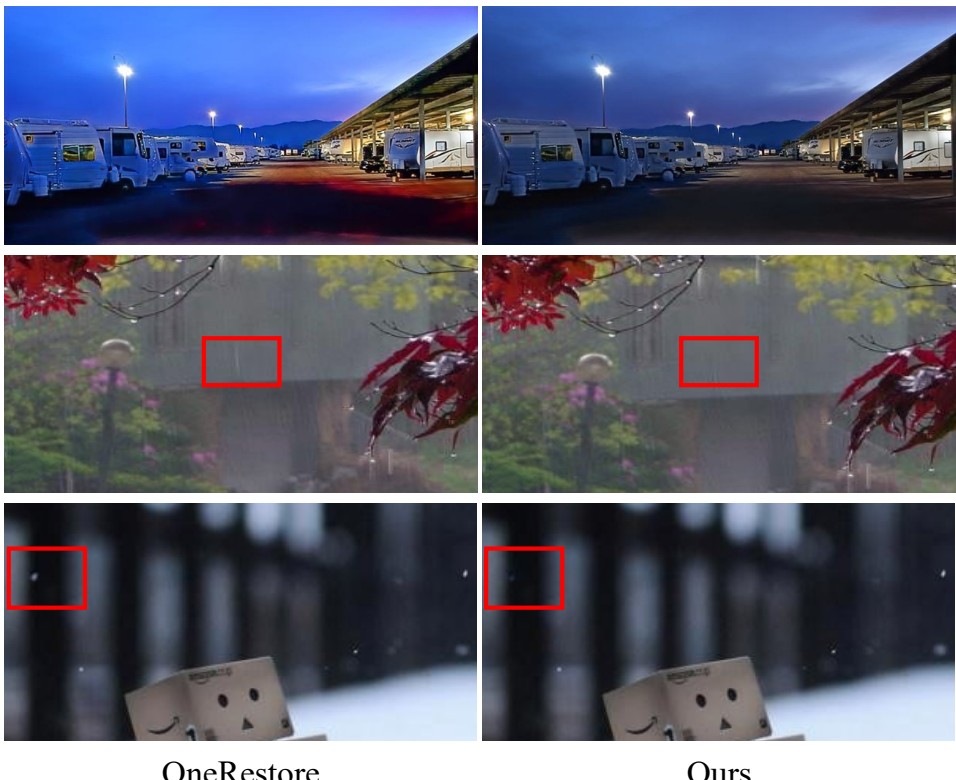

Figure 4: Qualitative Restoration Results on Real-World Datasets.

### G.3 SELECTIVE RESTORATION

Fig. 10 and Fig. 11 present qualitative results for selective restoration, where our model removes only the target degradation from composite images while preserving the others. Fig. 12 shows qualitative results for the same experiment conducted on the Blur-Noise-JPEG dataset. These results highlight the ability of our DisIR to selectively restore specific degradations while leaving the remaining degradations intact, in contrast to OneRestore, which struggles to provide such selective control. In Fig. 13, we further demonstrate that our model can simultaneously perform ratio control restoration and selective restoration on the same image, as indicated by the color of each arrow. Remarkably, this capability emerges even though the model was not explicitly supervised for such combined operations during training. These results underscore the fine-grained controllability of our approach in a variety of composite degradation scenarios.

### G.4 RESTORATION ORDER DEPENDENCY

Fig. 14 and Fig. 15 highlight the restoration order dependency observed in previous methods and demonstrate that our approach effectively mitigates this issue in composite degradation scenarios. As shown in these examples, our DisIR produces more consistent outputs regardless of the order in which degradations are removed, demonstrating strong order-invariant behavior. In contrast, OneRestore exhibits noticeable variations depending on the restoration sequence. These results visually confirm the effectiveness of our approach in disentangling and restoring individual degradations, regardless of the order in which they are removed. Furthermore, Fig. 15 presents qualitative results for the same experiment conducted on the Blur-Noise-JPEG dataset.

### G.5 IMAGE RESTORATION

We present qualitative results for general image restoration using our proposed CCDD-11 dataset. As discussed in the main paper, our DisIR achieves higher quantitative performance compared to previous approaches. In addition, qualitative results for $haze + snow$ restoration show that our DisIR removes the Snow component more effectively, as highlighted by the red box in Fig. 17. Fig. 18 presents the restoration of $low + haze + snow$ images. The red box highlights a challenging region where OneRestore struggles to produce satisfactory results, whereas our DisIR achieves noticeably better restoration performance. In Fig. 19, we show qualitative results for the restoration of Blur images. In the red box region, it can be seen that our DisIR produces better restoration quality compared to OneRestore. Our proposed method achieves performance improvements even without additional training on three composite images. We believe this indirectly suggests that our disentangle learning approach helps the model develop a deeper understanding of the distinct properties of each degradation, as well as composite degradations.

## H LARGE LANGUAGE MODEL USAGE

Following the conference rules about using Large Language Models (LLMs), we report how we used LLMs while writing this paper. LLMs were used only as basic writing help tools and did not help with research ideas, method creation, experiment planning, or data analysis.

Specifically, LLMs were used for the following purposes:

- **Grammar and Style Refinement**: Improving grammar, sentence structure, and ensuring consistency in academic writing style throughout the manuscript.
- **Logical Structure Enhancement**: Reorganizing sentence flow and improving the logical coherence of paragraphs to enhance readability.

All research concepts, methodological innovations, experimental designs, data collection, analysis, and scientific conclusions presented in this work are the result of the original research efforts of the authors. The core contributions, including the disentangled prompt learning framework, the four novel loss functions, and the experimental validation, were conceived and developed without the assistance of LLMs. The authors retain full responsibility for all technical content, claims, and conclusions presented in this paper.

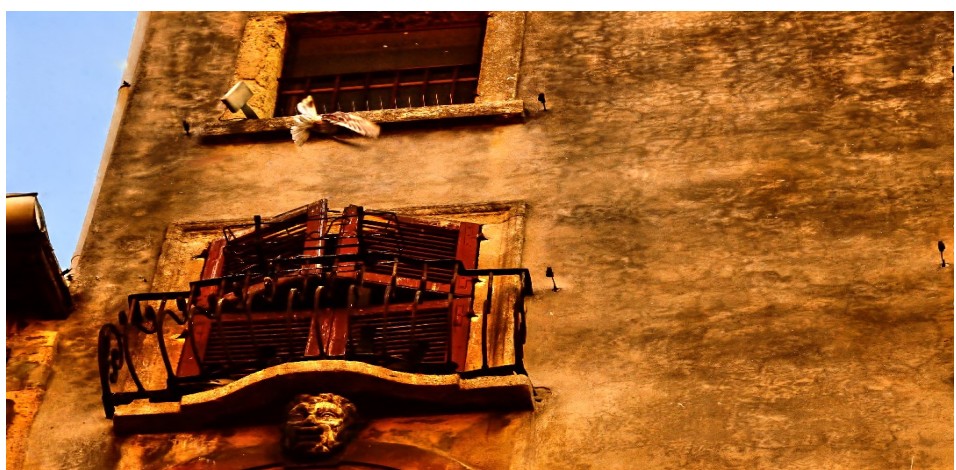

OneRestore (*clear* prompt)

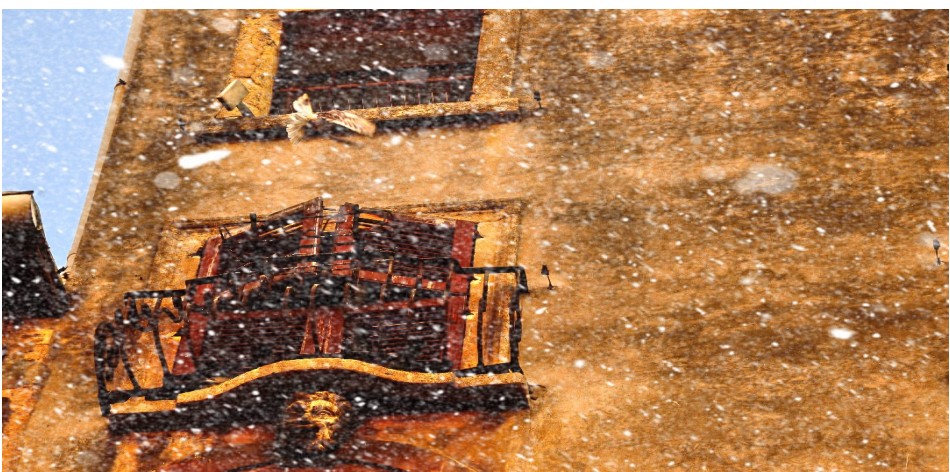

Ours (Identity Embedding)

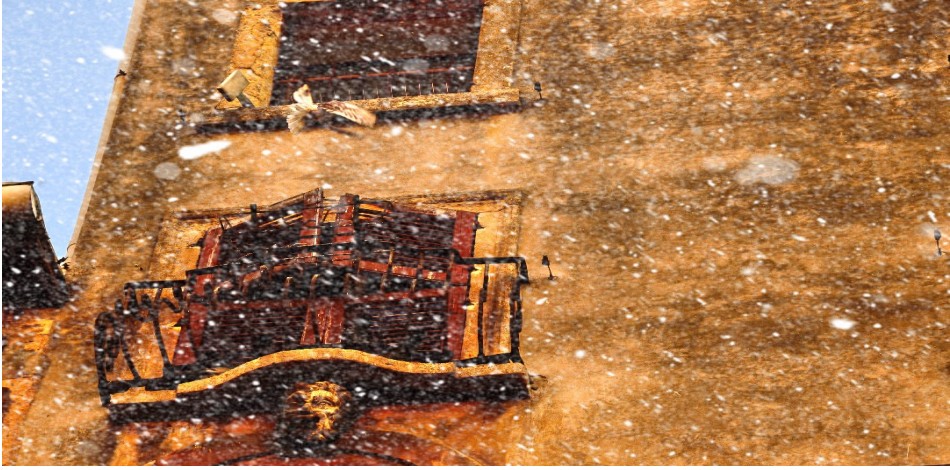

Snow (Target)

Figure 5: Qualitative comparison of identity operation for Snow image, where the Snow image serves as both the input and the target.

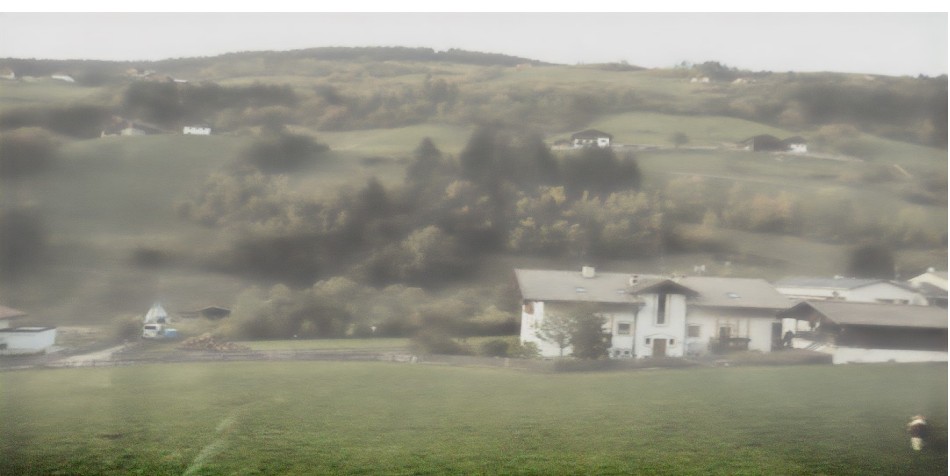

OneRestore (*clear* prompt)

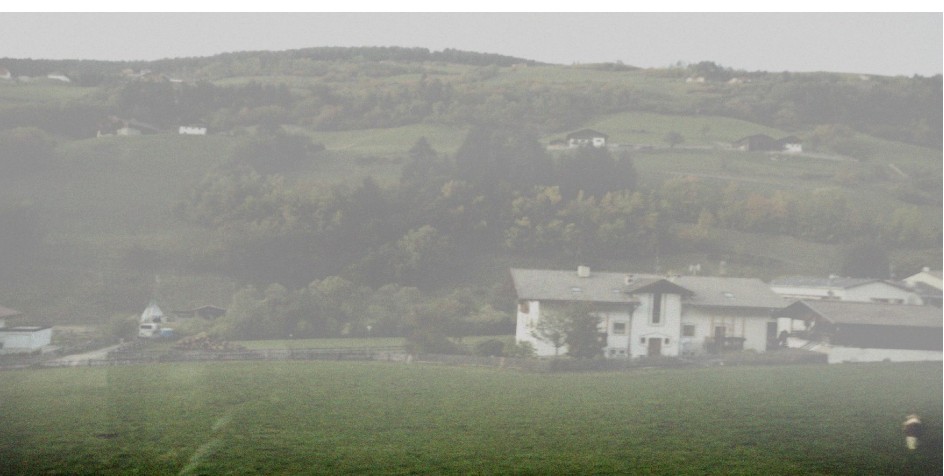

Ours (Identity Embedding)

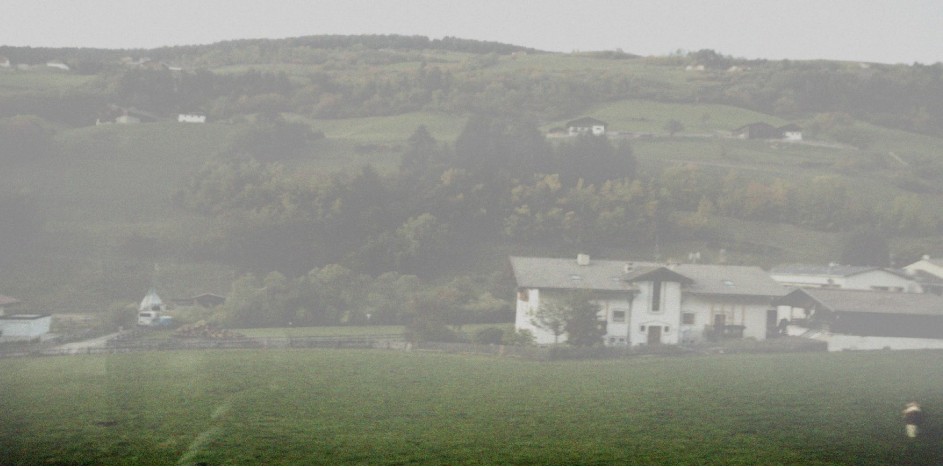

Low Haze (Target)

Figure 6: Qualitative comparison of identity operation for Low+Haze image, where the Low+Haze image serves as both the input and the target.

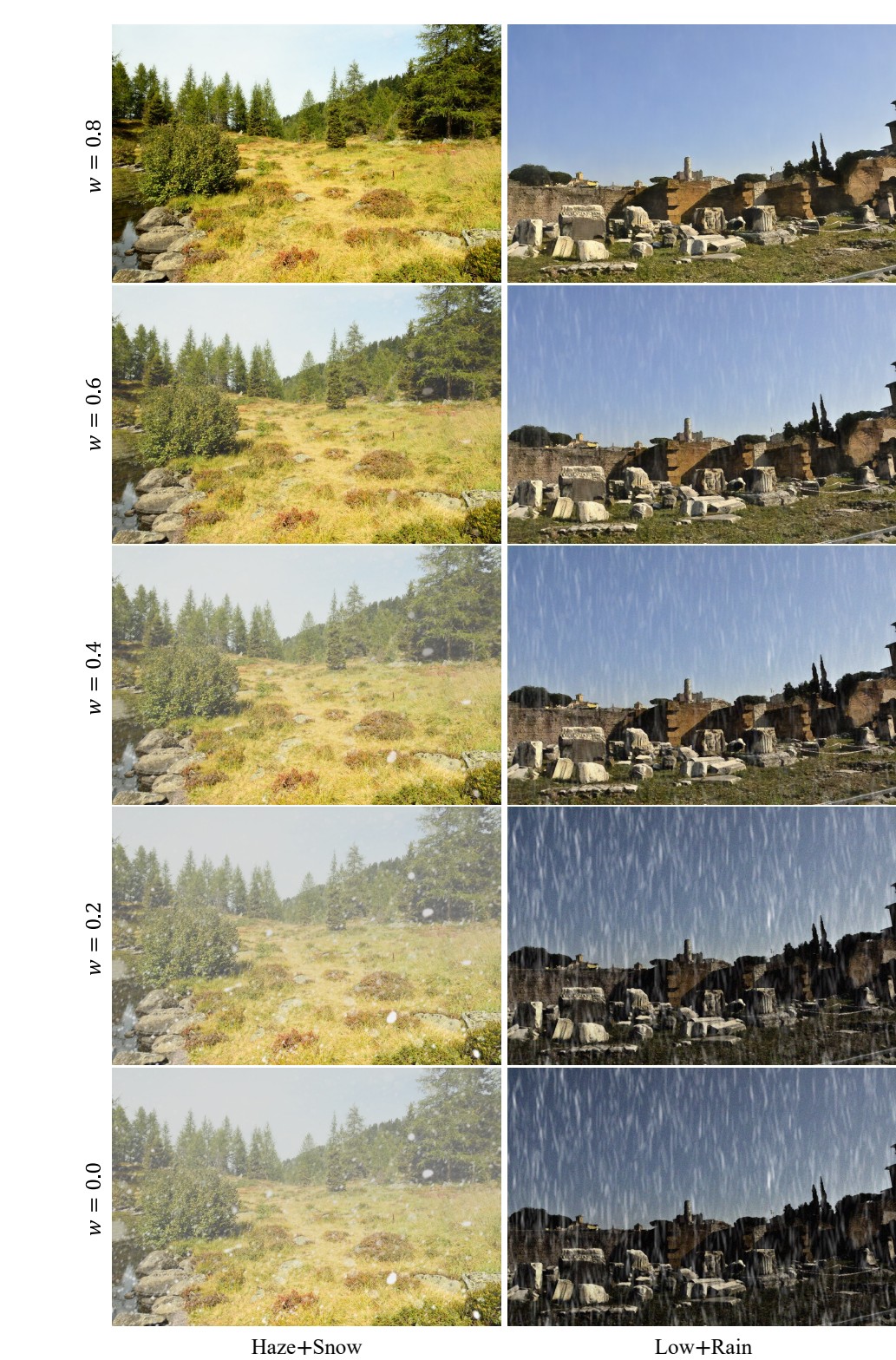

Figure 7: Qualitative results of Ratio Control Embedding with varying $w$ values for Haze+Snow and Low+Rain.

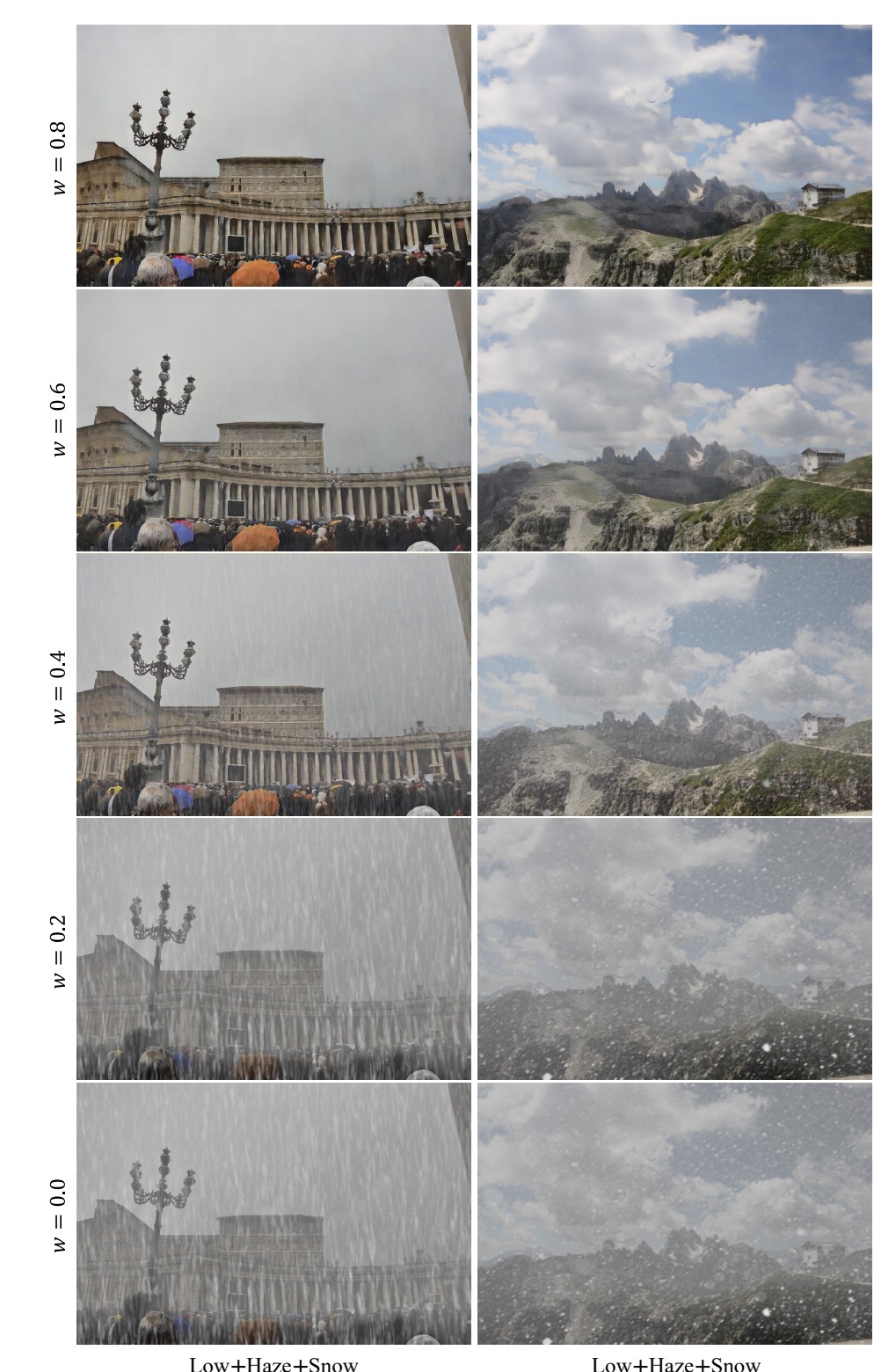

Figure 8: Qualitative results of Ratio Control Embedding with varying $w$ values for Low+Haze+Snow and Low+Haze+Rain.

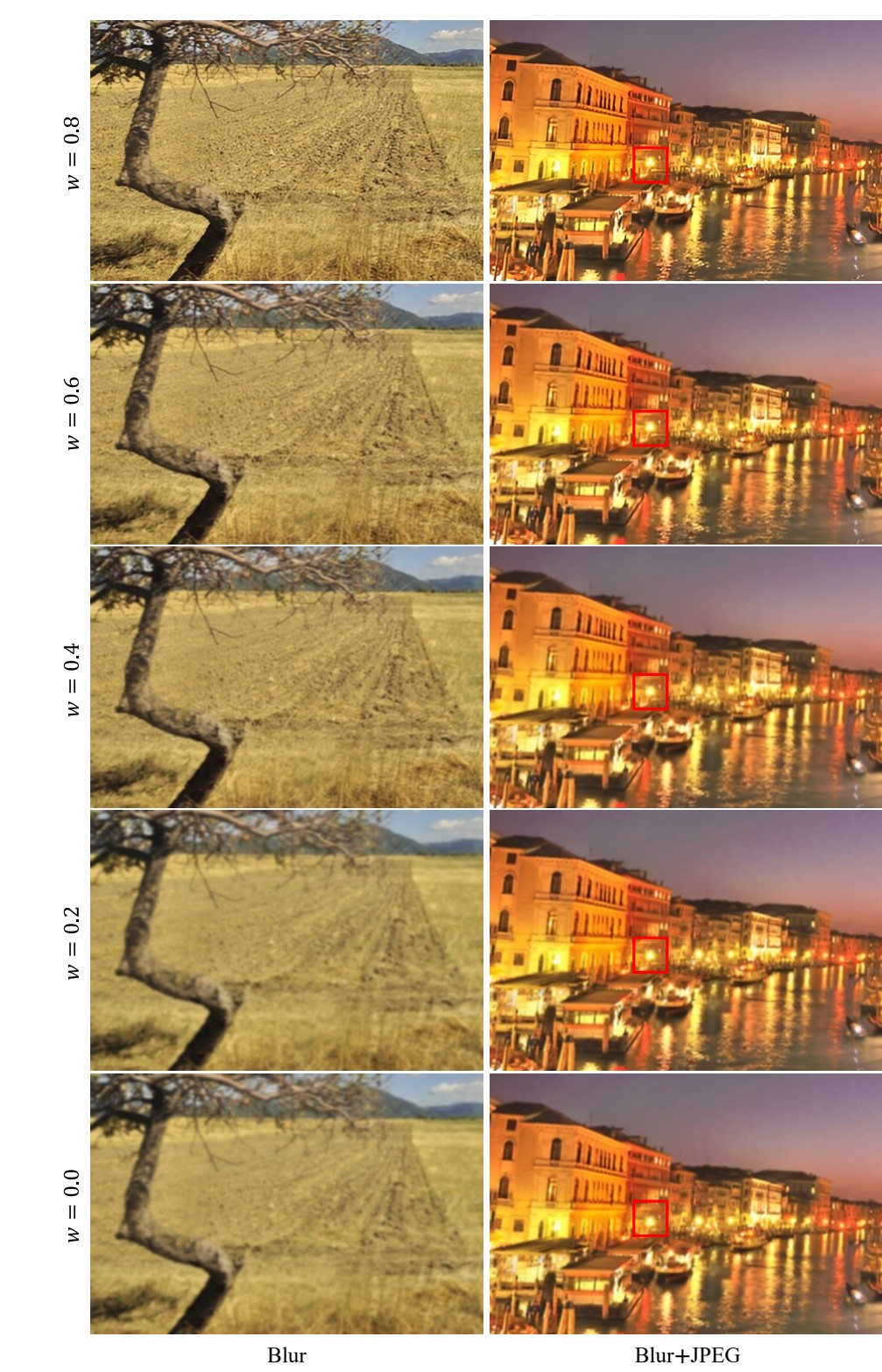

Figure 9: Qualitative results of Ratio Control Embedding with varying $w$ values for Blur and Blur+JPEG.

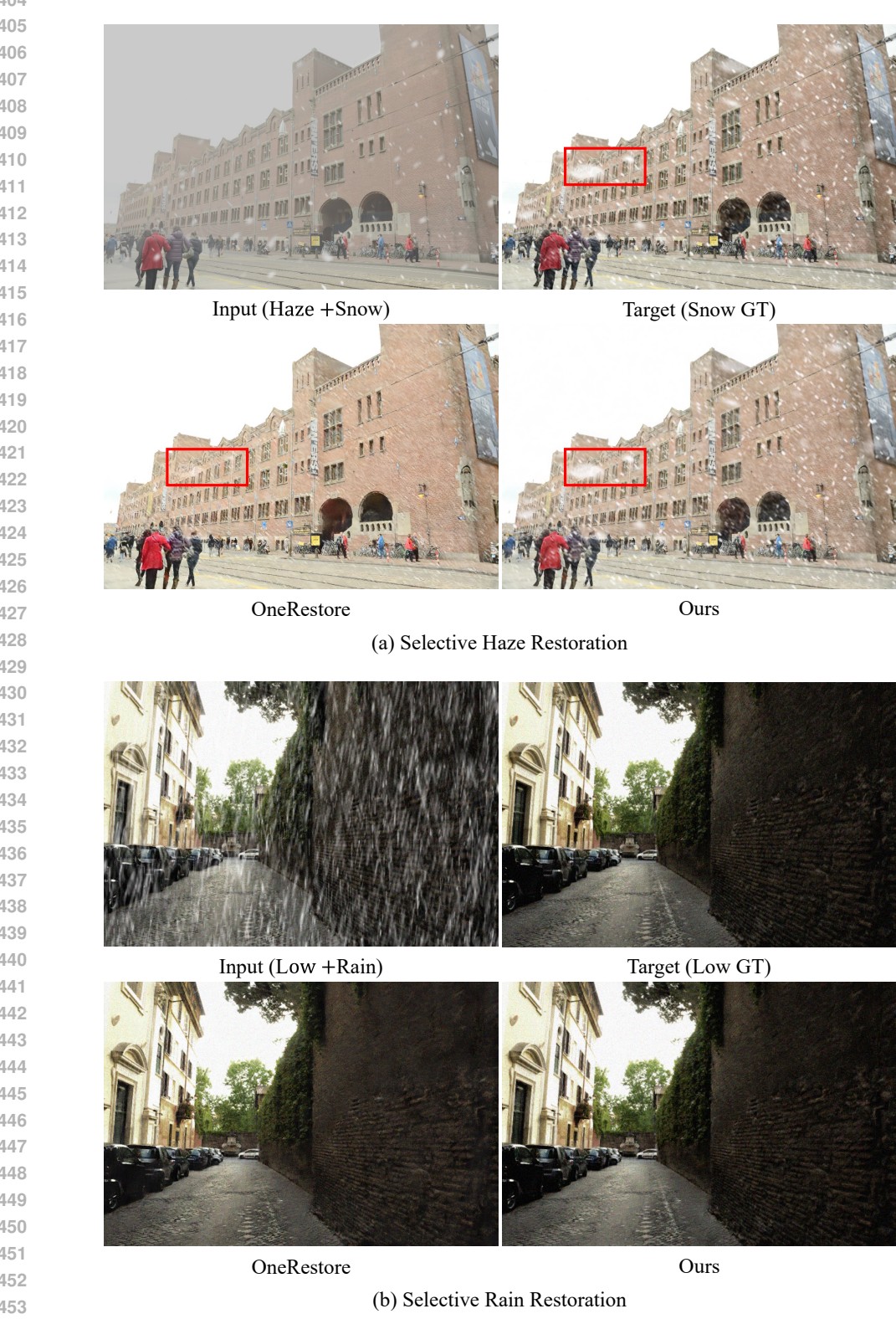

(a) Selective Haze Restoration

(b) Selective Rain Restoration

Figure 10: (a) Qualitative results of selective restoration for the Haze component in Haze+Snow images. (b) Qualitative results of selective restoration for the Rain component in Low+Rain images.

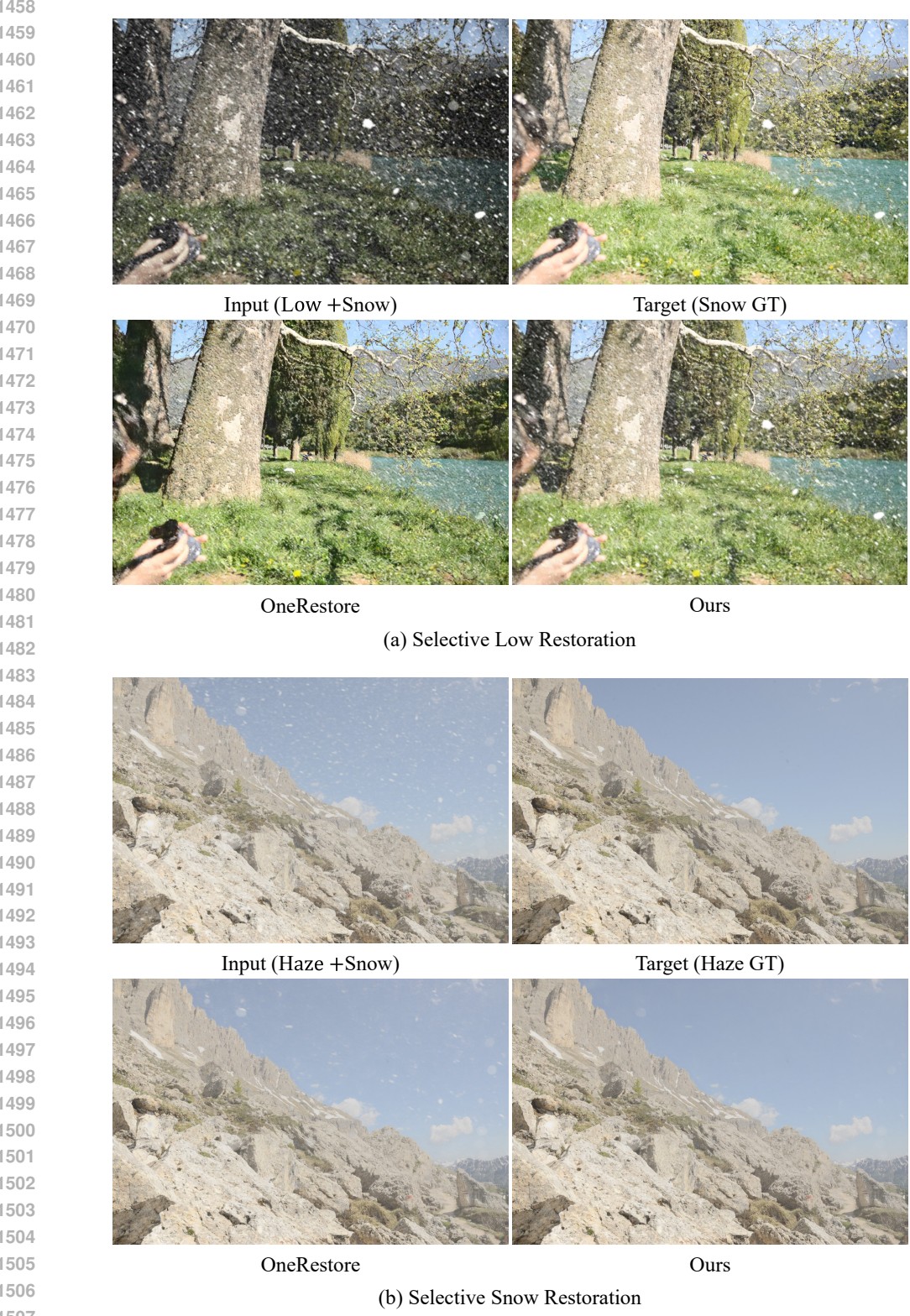

(a) Selective Low Restoration

(b) Selective Snow Restoration

Figure 11: (a) Qualitative results of selective restoration for the Low component in Low+Snow images. (b) Qualitative results of selective restoration for the Snow component in Haze+Snow images.

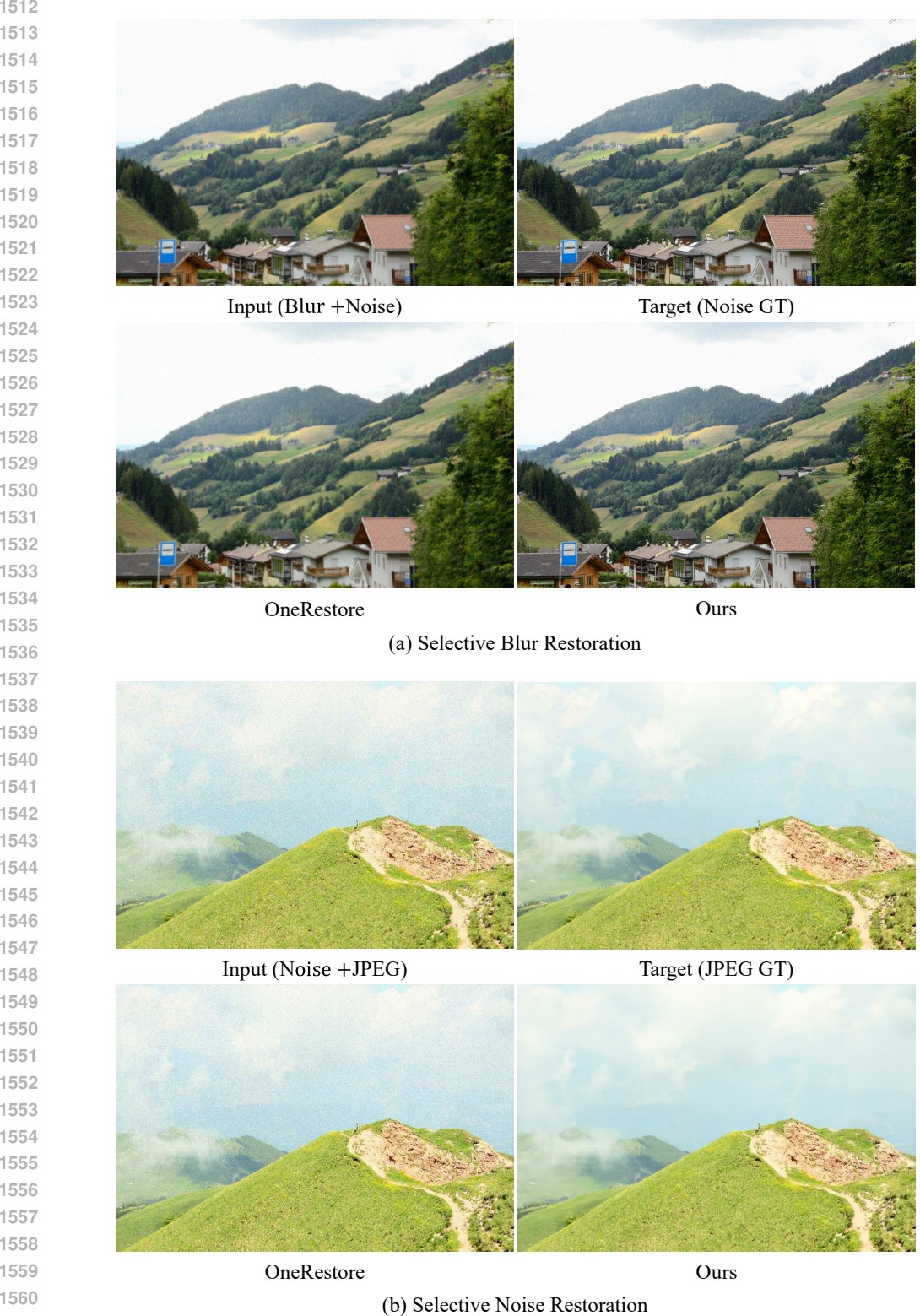

(a) Selective Blur Restoration

(b) Selective Noise Restoration

Figure 12: (a) Qualitative results of selective restoration for the Low component in Blur+Noise images. (b) Qualitative results of selective restoration for the Snow component in Noise+JPEG images.

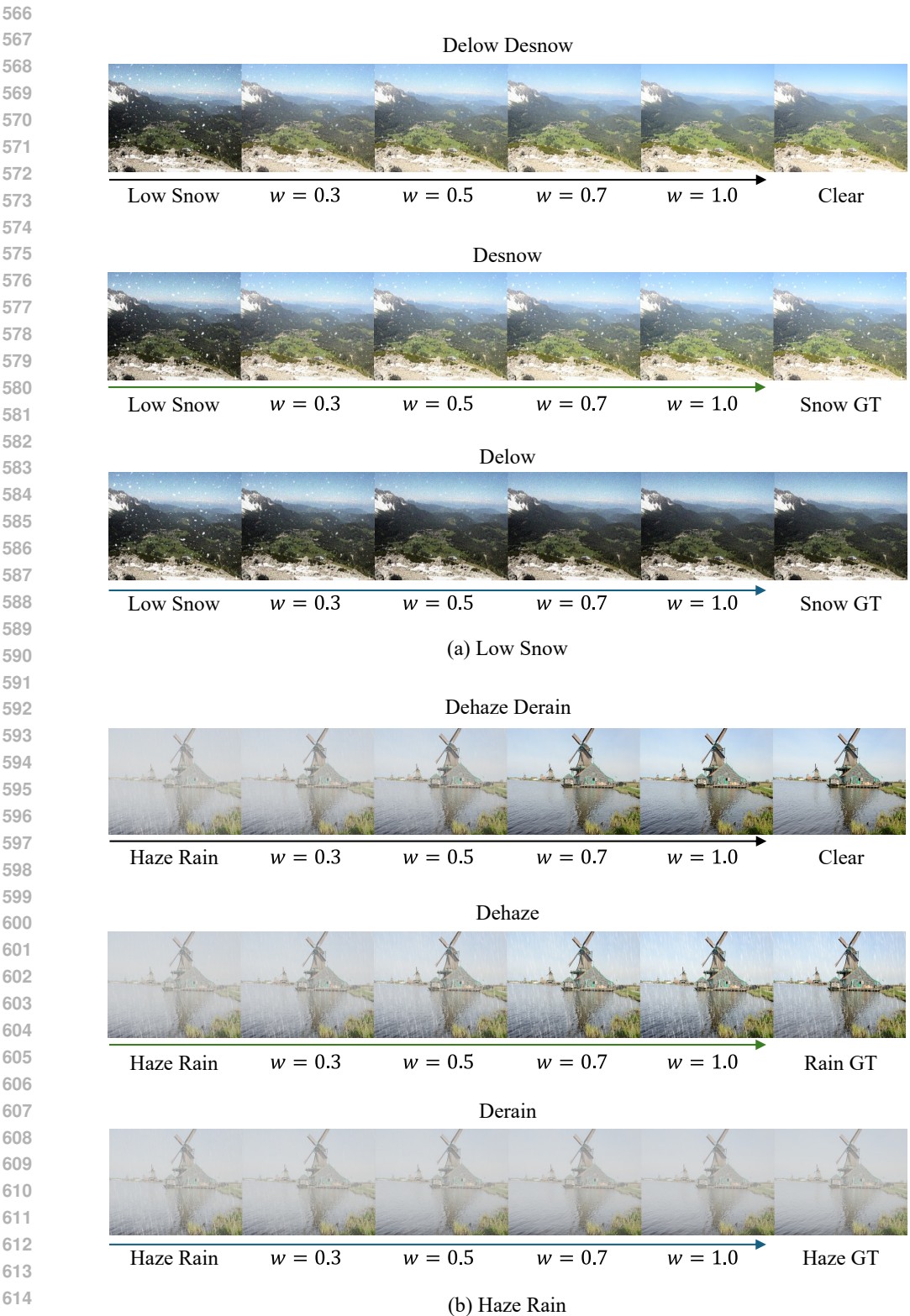

Figure 13: Example results of simultaneous Ratio Control Restoration and Selective Restoration. For each image, different text embeddings and ratio control values are applied, as indicated by the color of each arrow.

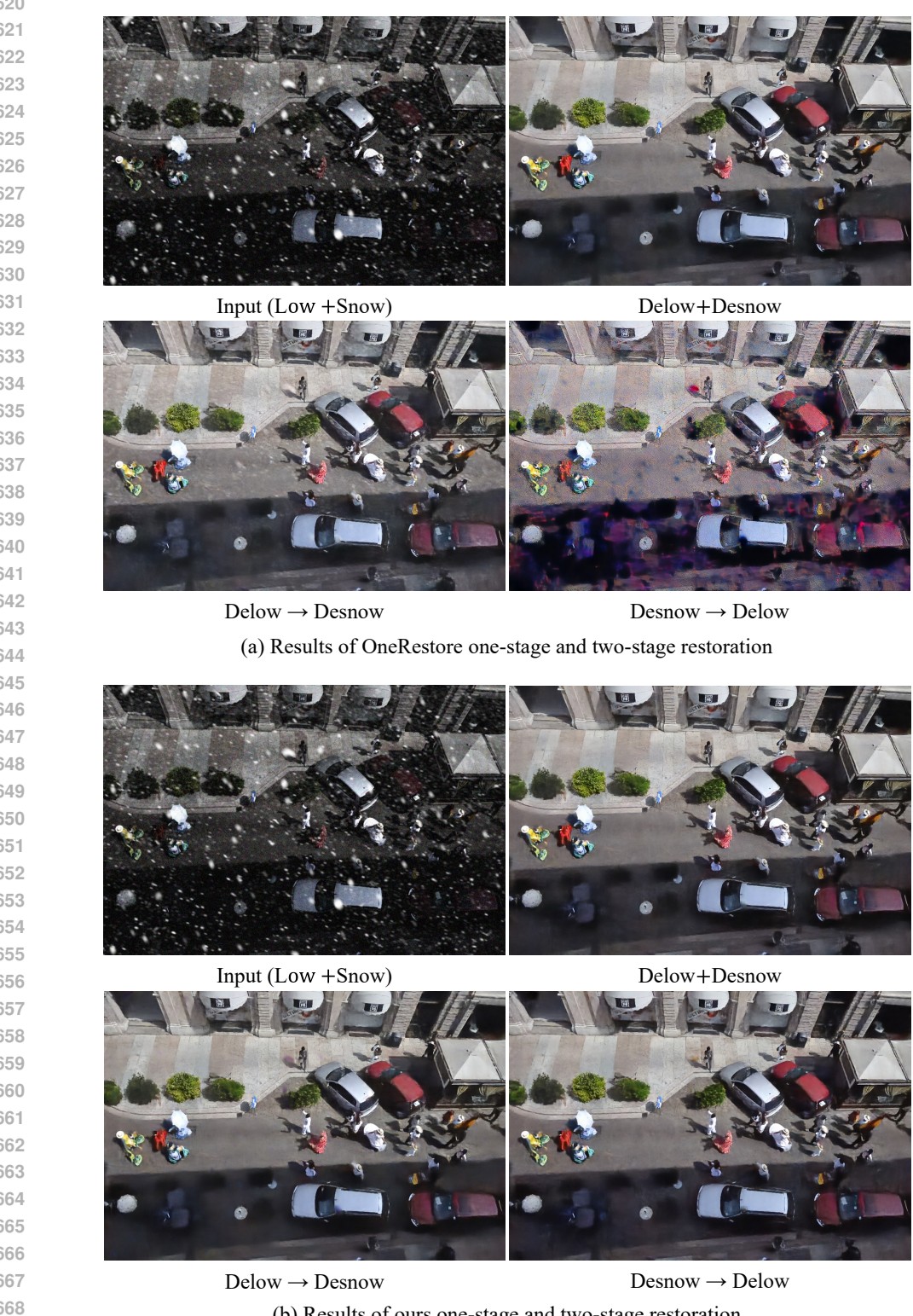

Input (Low +Snow)    Delow+Desnow

Delow → Desnow    Desnow → Delow

(a) Results of OneRestore one-stage and two-stage restoration

Input (Low +Snow)    Delow+Desnow

Delow → Desnow    Desnow → Delow

(b) Results of ours one-stage and two-stage restoration

Figure 14: Comparison of one-stage and two-stage restoration results for (a) OneRestore and (b) Ours on Low+Snow images, illustrating the differences between single-stage and two-stage restoration approaches.

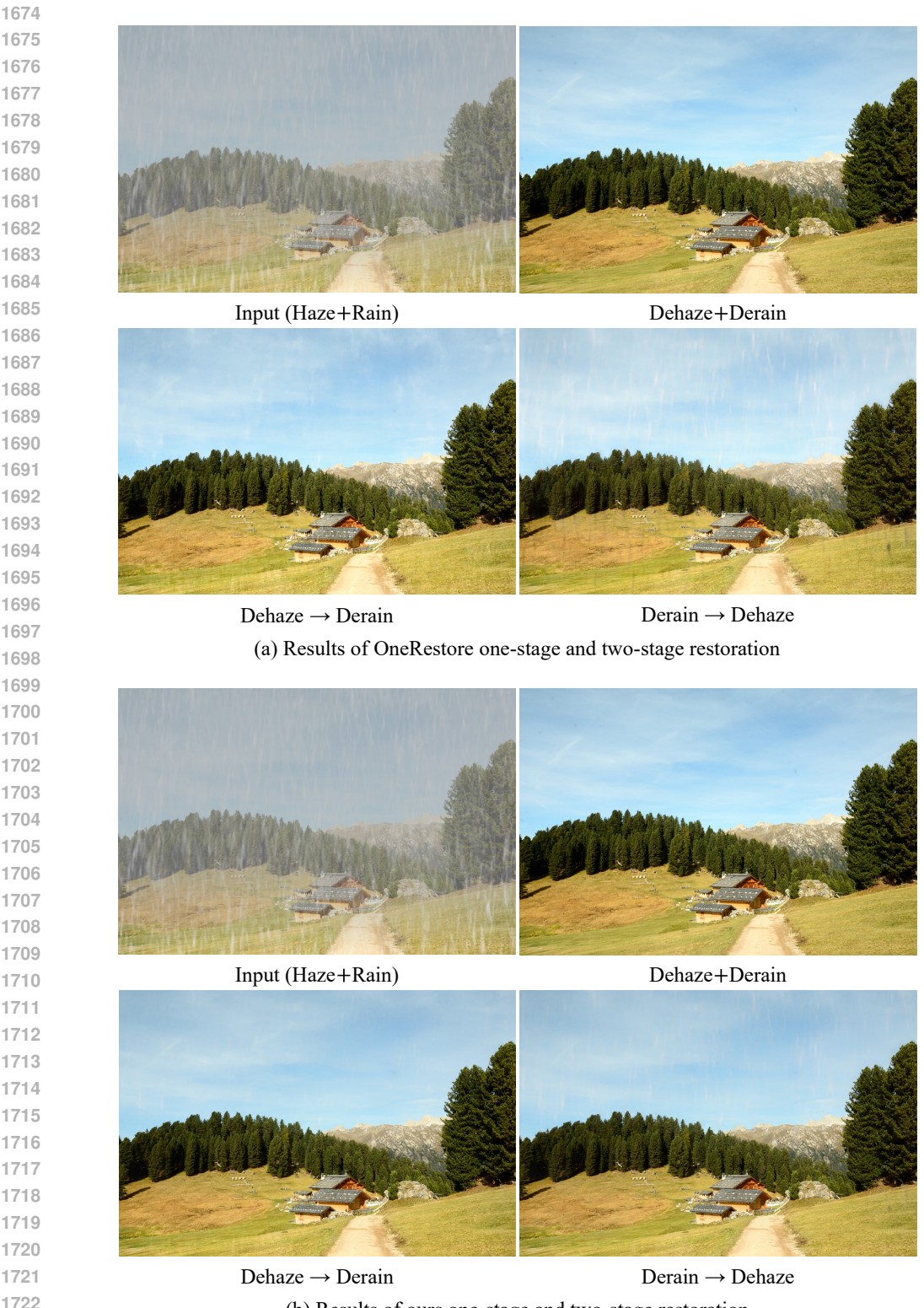

Input (Haze+Rain)             Dehaze+Derain

Dehaze → Derain             Derain → Dehaze

(a) Results of OneRestore one-stage and two-stage restoration

Input (Haze+Rain)             Dehaze+Derain

Dehaze → Derain             Derain → Dehaze

(b) Results of ours one-stage and two-stage restoration

Figure 15: Comparison of one-stage and two-stage restoration results for (a) OneRestore and (b) Ours on Haze+Rain images, illustrating the differences between single-stage and two-stage restoration approaches.

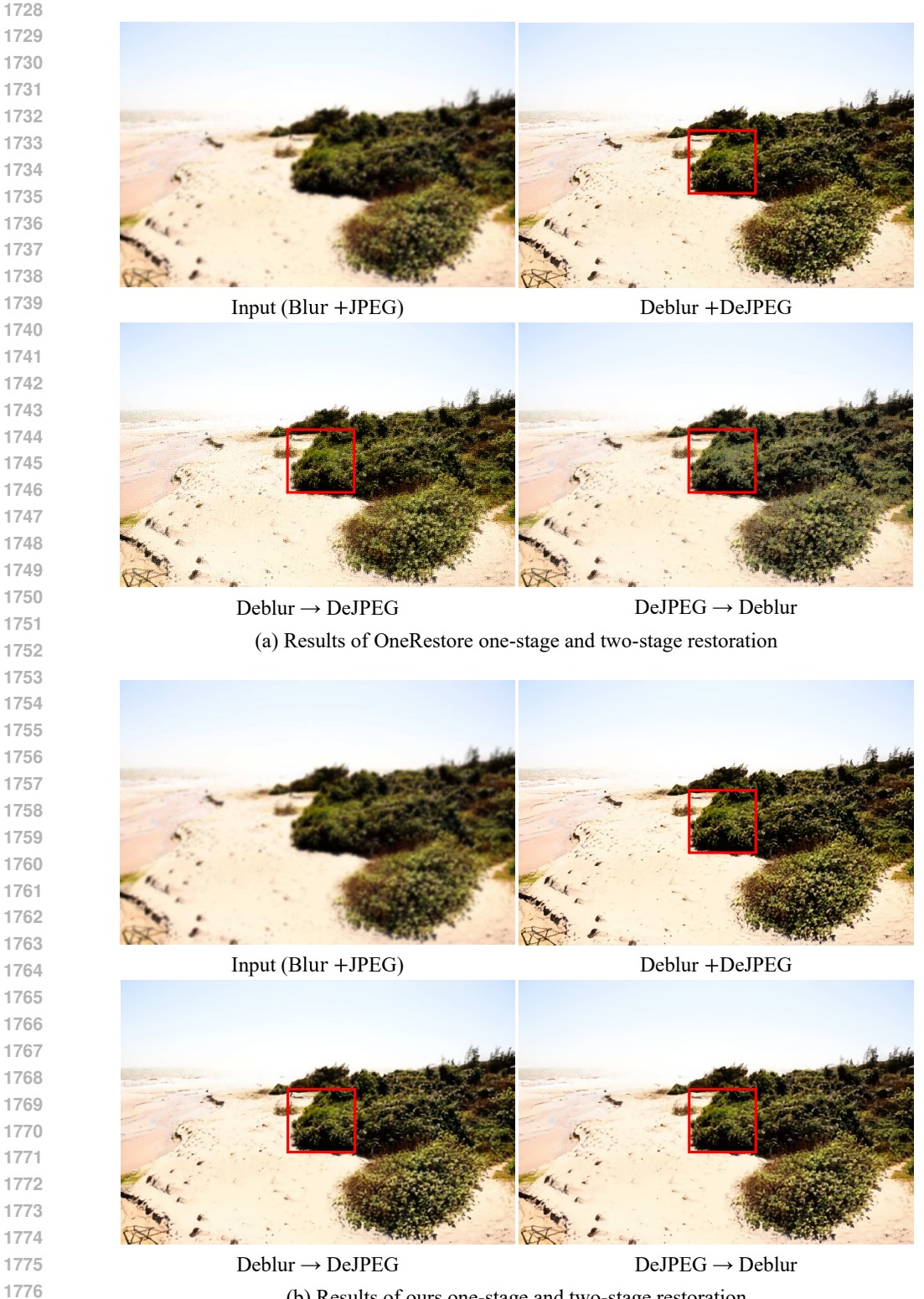

Figure 16: Comparison of one-stage and two-stage restoration results for (a) OneRestore and (b) Ours on Blur+JPEG images, illustrating the differences between single-stage and two-stage restoration approaches.

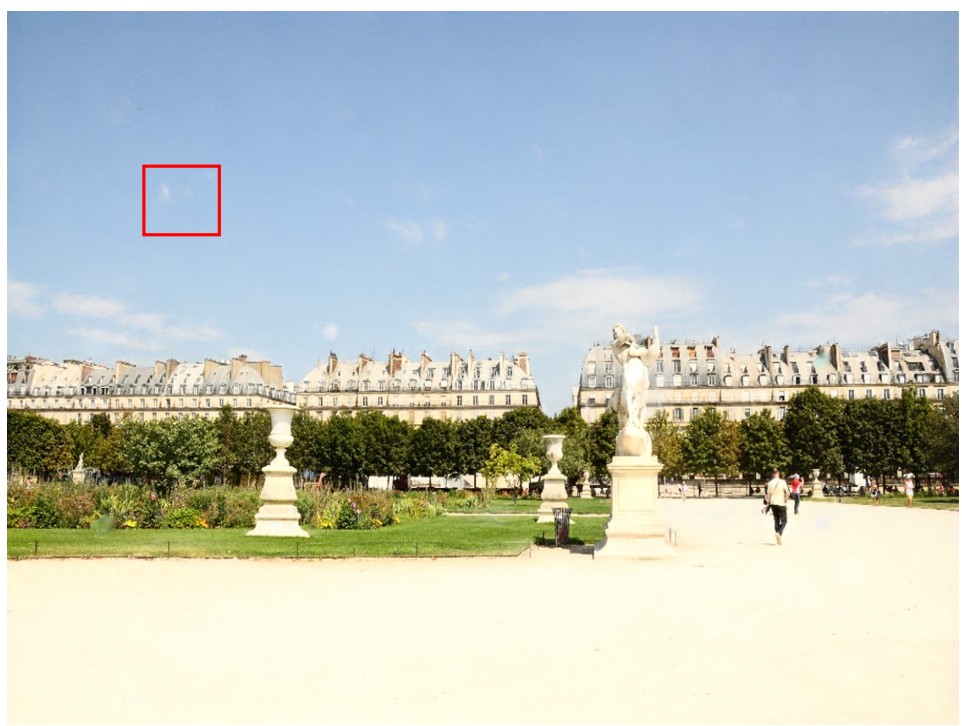

OneRestore

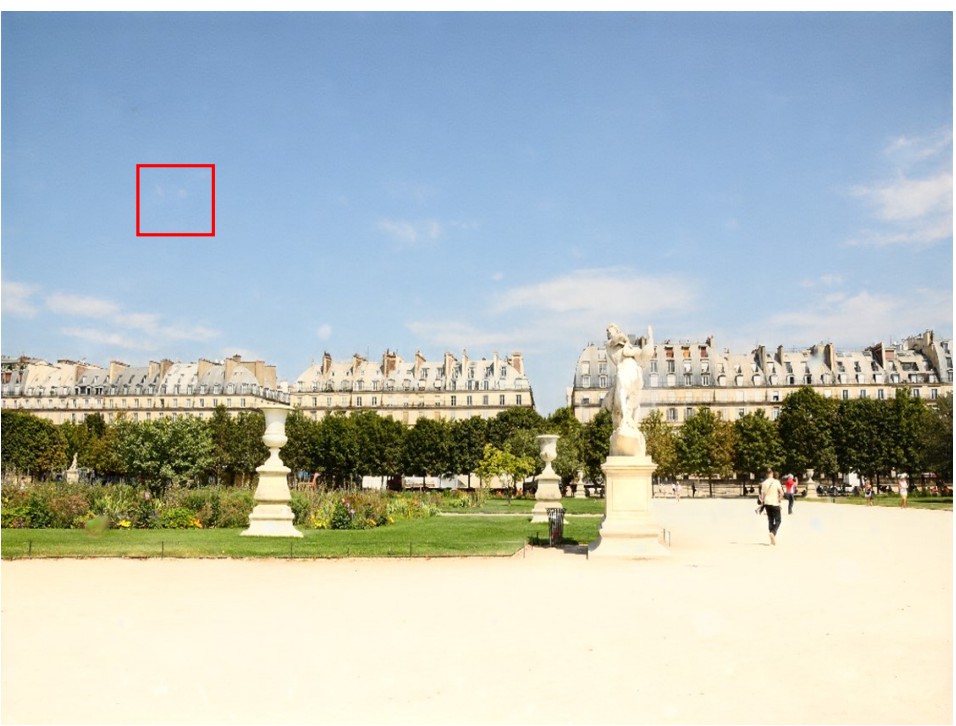

Ours

Figure 17: Qualitative comparison of Haze+Snow image restoration results between OneRestore and Ours.

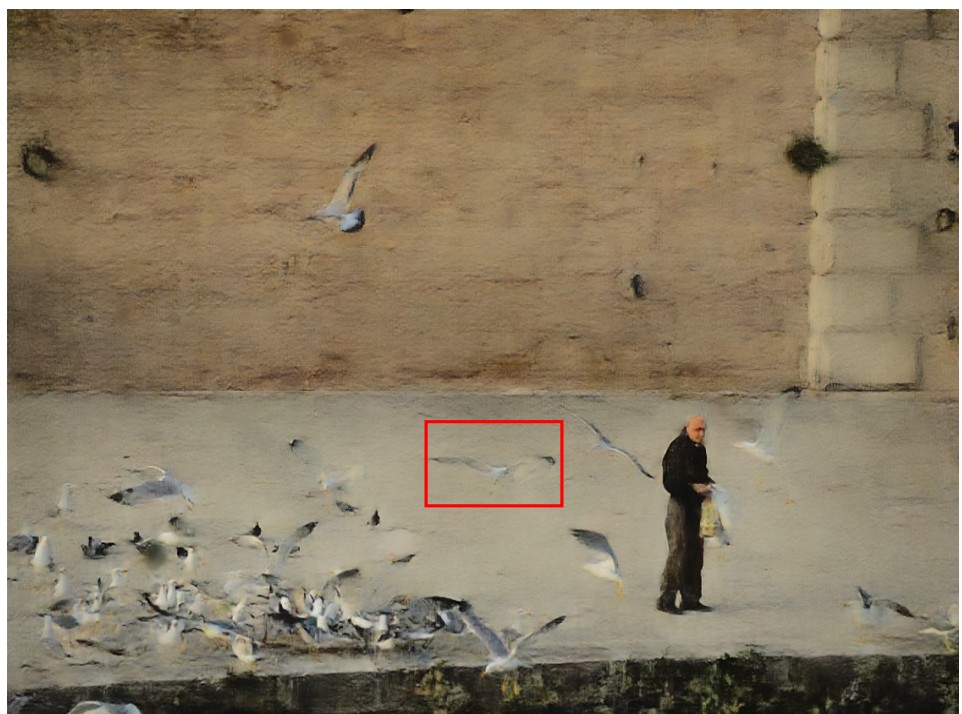

OneRestore

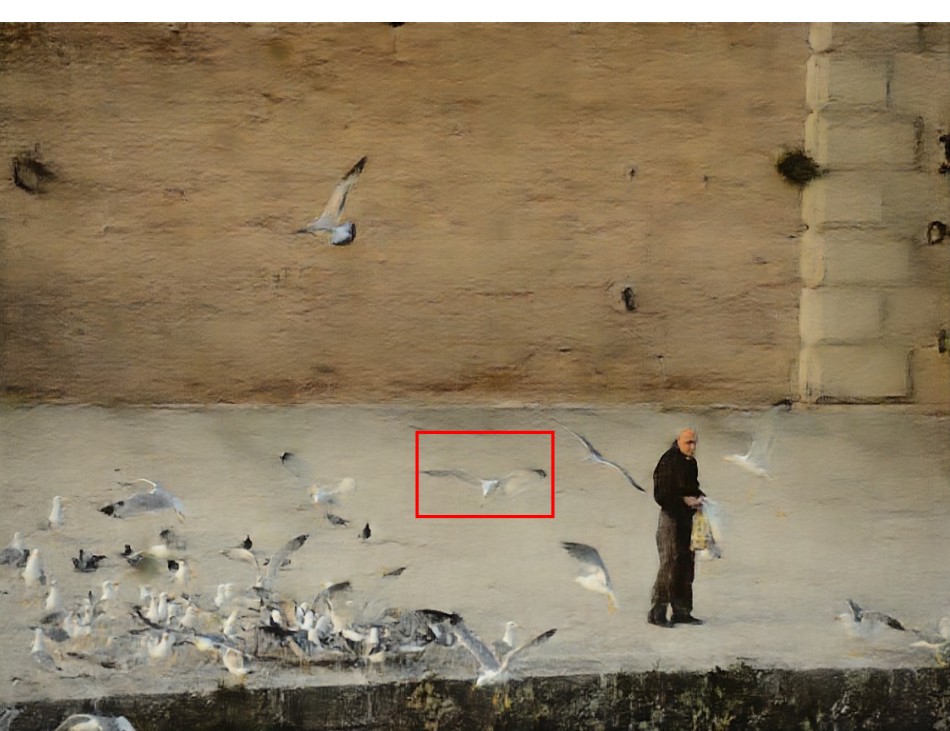

Ours

Figure 18: Qualitative comparison of Low+Haze+Snow image restoration results between OneRestore and Ours.

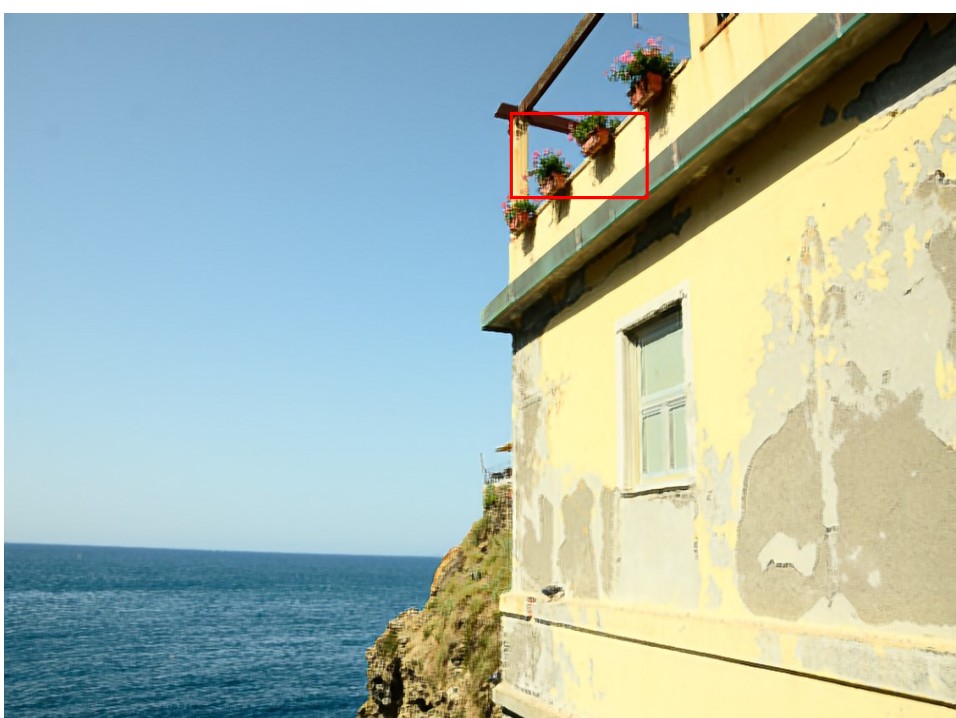

OneRestore

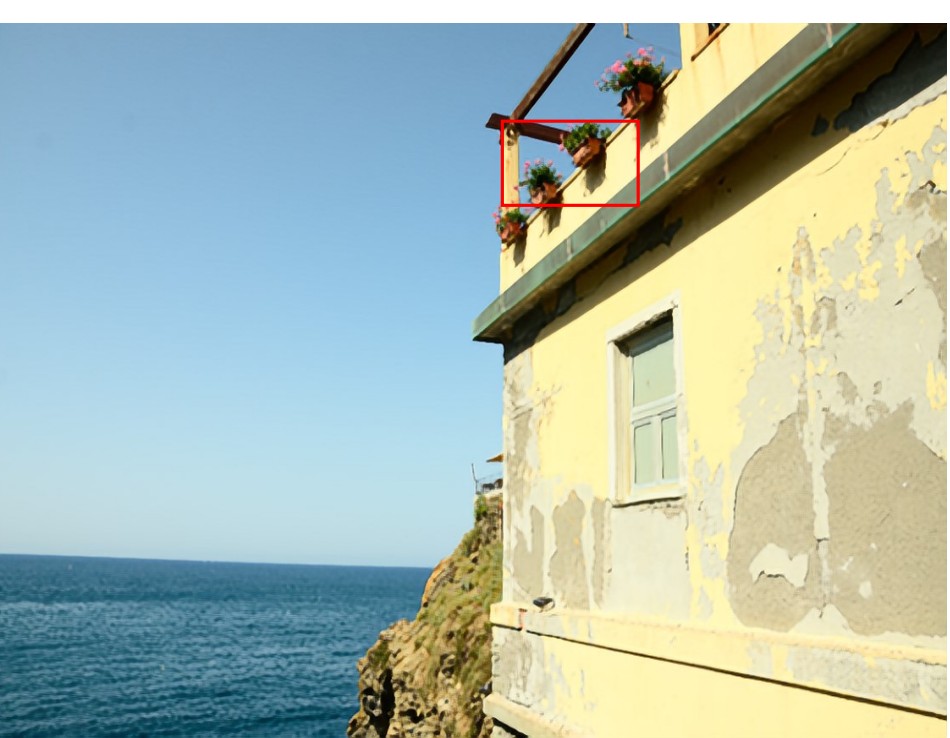

Ours

Figure 19: Qualitative comparison of Blur image restoration results between OneRestore and Ours.

