# OpenReview forum: "DisIR: Disentangled Learning of Controllable All-in-One Image Restoration under Composite Degradations"
_ICLR.cc/2026/Conference — ICLR 2026 Conference Withdrawn Submission_

### Official Review · Reviewer_sL2r · 2025-10-19

**Soundness:** 2
**Presentation:** 2
**Contribution:** 2
**Rating:** 4
**Confidence:** 4

**Summary:**

This paper tackles the problem of restoring images corrupted by multiple overlapping degradations (e.g., haze and snow). The authors argue that while existing "all-in-one" models can handle such composite inputs, they lack fine-grained controllability. They propose DisIR, a novel framework that introduces a disentangled learning pipeline to enable precise control over the restoration process. The core of their method is a set of four specialized loss functions designed to guide the model to: preserve the original image when requested, remove degradations at a controllable intensity, selectively target specific degradations while leaving others untouched, and produce consistent results regardless of the processing order. The proposed method is evaluated on a new benchmark, CCDD-11, and is shown to outperform existing approaches, offering a solution towards more flexible and user-directed image restoration systems.

**Strengths:**

* The concept of the "ratio control embedding" – a linear interpolation between identity and degradation embeddings – is an elegant and intuitive solution for achieving continuous control over restoration intensity. This is a notable technical contribution.
* The paper provides extensive comparative experiments on the proposed CCDD-11 dataset, demonstrating superior overall performance against a wide range of baselines.

**Weaknesses:**

* My major concern lies in the ablation study. It fails to isolate the contribution of each component (e.g., the Identity loss is active in all rows). and relies solely on final PSNR on a single dataset, providing no dedicated metrics to quantify the claimed controllability (e.g., accuracy of selective removal, linearity of ratio control).
* There is no analysis of the model's generalization to real-world data or other degradation benchmarks, leaving the robustness and broader applicability of the method in question.
* The description of the proposed method in Section 3 is disjointed and lacks a coherent narrative. The core concept of "disentangled learning" is not formally defined or motivated. The presentation jumps between architectural components (embeddings) and training objectives (loss functions) without a clear, unifying framework. This makes it difficult to discern the fundamental algorithmic contribution beyond the introduction of several additive components (new embeddings and new losses) to a strong baseline.

**Questions:**

See weaknesses

---

### Official Review · Reviewer_Ms95 · 2025-10-31

**Soundness:** 1
**Presentation:** 2
**Contribution:** 2
**Rating:** 2
**Confidence:** 5

**Summary:**

This paper proposes DisIR to address mixed degradation restoration, especially in cases of complex mixtures of multiple degradation types, by selectively processing individual degradation types. However, the next task seems to lack practical applications: no one needs to intentionally preserve other degradations when restoring low-quality images.

Furthermore, although this paper claims that DisIR can be integrated into a controllable architecture without redesigning its architecture, it has only been validated on one architecture (OneRestore), failing to demonstrate its compatibility with traditional image restoration networks (SwinIR, NAFNet, Restormer) or current diffusion-based ones.

Finally, while this paper claims to solve the complex degradation problem, validation was only performed on mixed weather datasets, not on broader scenarios such as low-light + blur + noise, downsampling + blur + noise + compression, etc.

Overall, I believe this paper has significant room for improvement and is not suitable for publication now.

**Strengths:**

1. This paper is well-structured.
2. The proposed method is novel and well-developed. However, due to insufficient experiments, its transferability and effectiveness have not been strongly verified.

**Weaknesses:**

1. The existing work is not well reviewed. Lines 54-55 state that "most all-in-one restoration studies have focused on treating the types of degradation individually." This might have been true a year ago. However, nowadays, many works address mixed degradation restoration, such as DCPT [1] and MoceIR [2], have been validated on CDD. UniRes [3], for example, addresses a broader range of hybrid degradation problems. Similarly, in the related work section, this paper only lists work from 2024 onwards, ignoring a significant amount of existing work, such as VLUNet [4] and UniRestore [5].
2. This paper lacks sufficient experiments.
- This paper claims that DisIR applies to multiple architectures, but the experimental results do not reflect this, as it was only tested on the OneRestore architecture.
- This paper claims that DisIR can solve the problem of mixed degradation, but the experimental results do not reflect this, as it was only tested on mixed weather datasets, and its effectiveness on a wider range of mixed degradation (e.g., low-light + blur + noise LoL-Blur [6] dataset, downsampling + blur + noise + compression Real-world Super-resolution) has not been verified.
- The visual improvements given in the appendix are not significant enough.

[1] Universal Image Restoration Pre-training via Degradation Classification. ICLR 2025.

[2] Complexity Experts are Task-Discriminative Learners for Any Image Restoration. CVPR 2025.

[3] Universal Image Restoration for Complex Degradations. ICCV 2025.

[4] Vision-Language Gradient Descent-driven All-in-One Deep Unfolding Networks. CVPR 2025.

[5] UniRestore: Unified Perceptual and Task-Oriented Image Restoration Model Using Diffusion Prior. CVPR 2025.

[6] LEDNet: Joint Low-light Enhancement and Deblurring in the Dark. ECCV 2022.

**Questions:**

1. On the CDD dataset, it should also be compared with state-of-the-art methods [1,2]. However, this paper shows a significant performance gap compared to MoceIR and DCPT. Is this due to limitations in the number of parameters?

[1] Universal Image Restoration Pre-training via Degradation Classification. ICLR 2025.

[2] Complexity Experts are Task-Discriminative Learners for Any Image Restoration. CVPR 2025.

---

### Official Review · Reviewer_2mxA · 2025-11-02

**Soundness:** 2
**Presentation:** 2
**Contribution:** 2
**Rating:** 2
**Confidence:** 5

**Summary:**

The paper introduces DisIR, a new training framework that turns an existing all-in-one restoration network into a controllable system for composite degradations.

**Strengths:**

1. This work applies continuous intensity control and selective single-degradation removal in an all-in-one model without architectural redesign.

**Weaknesses:**

1. The novelty of this work is not so significant. All technical contributions reside in loss functions; the encoder–decoder and embedder are imported unchanged from OneRestore. Discuss whether the same losses can be layer-wise or attention-head-wise regularisers inside the SDTB block, or provide a theoretical justification that architectural change is unnecessary.
2. Paper concedes that training becomes “impractical” for triple-composite images; yet real photos can contain five or more degradations (blur, noise, JPEG, haze, rain, low-light, glare, etc.). Report compute/memory growth w.r.t. number of degradations d (approx. O(d²) pairs for permutations). Evaluate on a 5-degradation subset or discuss hierarchical or recursive application of DisIR.
3. The latest related work was published in 2024. The reviews don't contain the all-in-one work published in 2025. As far as I know, there are many excellent works in 2025. Additionally, the experiments should contain these methods.

**Questions:**

Please refer to the weaknesses.

**Details Of Ethics Concerns:**

None.

---

### Note · Authors · 2025-11-14

I have read and agree with the venue's withdrawal policy on behalf of myself and my co-authors.